

# Assessing the benefits of Imaging Infrared Radiometer observations to the CALIOP version 4 cloud and aerosol discrimination algorithm

Thibault Vaillant de Guélis[1,2,3], Gérard Ancellet[1], Anne Garnier[2,3], Laurent C.-Labonnote[4], Jacques Pelon[1], Mark A. Vaughan[3], Zhaoyan Liu[3], and David M. Winker[3]

[1]LATMOS/IPSL, CNRS, Sorbonne Université, UVSQ, 75252 Paris, France
[2]Science Systems and Applications, Inc., Hampton, VA 23666, USA
[3]NASA Langley Research Center, Hampton, VA 23681, USA
[4]LOA, Université de Lille, 59655 Villeneuve d'Ascq, France

**Correspondence:** Thibault Vaillant de Guélis (thibault.vaillantdeguelis@outlook.com)

**Abstract.** The features detected in monolayer atmospheric columns sounded by the Cloud and Aerosol Lidar with Orthogonal Polarization (CALIOP) and classified as cloud or aerosol layers by the CALIOP version 4 (V4) cloud and aerosol discrimination (CAD) algorithm are reassessed using perfectly collocated brightness temperatures measured by the Imaging Infrared Radiometer (IIR) onboard the same satellite. Using the IIR's three wavelength measurements of layers that are confidently

classified by the CALIOP CAD algorithm, we calculate two-dimensional (2-D) probability distribution functions (PDFs) of IIR brightness temperature differences (BTDs) for different cloud and aerosol types. We then compare these PDFs with 1-D radiative transfer simulations for ice and water clouds and dust and marine aerosols. Using these IIR 2-D BTD signature PDFs, we develop and deploy a new IIR-based CAD algorithm and compare the classifications obtained to the results reported by the CALIOP-only V4 CAD algorithm. IIR observations are shown to be able to identify clouds with a good accuracy. The IIR cloud

identifications agree very well with layers classified as confident clouds by the V4 CAD algorithm (88 %). More importantly, simultaneous use of IIR information reduces the ambiguity in a notable fraction of "not confident" V4 cloud classifications. 28 % and 14 % of the ambiguous V4 cloud classifications are confirmed thanks to the IIR observations in the tropics and in the midlatitudes respectively. IIR observations are of relatively little help in deriving high confidence classifications for most aerosols, as the low altitudes and small optical depths of aerosol layers yield IIR signatures that are similar to those from clear

skies. However, misclassifications of aerosol layers, such as dense dust or elevated smoke layers, by the V4 CAD algorithm can be corrected to cloud layer classification by including IIR information. 10 %, 16 %, and 6 % of the ambiguous V4 dust, polluted dust, and tropospheric elevated smoke respectively are found to be misclassified cloud layers by the IIR measurements.



## 1 Introduction

Since its launch in 2006, the Cloud-Aerosol Lidar and Infrared Pathfinder Satellite Operations (CALIPSO) mission (Winker et al., 2010) has provided vertically-resolved measurements of aerosols and clouds between 81.8° S and 81.8° N thanks to its primary instrument: the two-wavelength (532 and 1064 nm) Cloud and Aerosol Lidar with Orthogonal Polarization (CALIOP).

Discrimination between cloud and aerosol layers relies on the combined analysis of several carefully calibrated quantities (Getzewich et al., 2018; Kar et al., 2018; Vaughan et al., 2019). In the version 4 (V4) data products release, the cloud and

aerosol discrimination (CAD) algorithm (Liu et al., 2019) uses five-dimensional probability distribution functions (PDFs) where dimensions are the layer-mean attenuated backscatter at 532 nm, the layer-mean total attenuated color ratio (mean at 1064 nm divided by the mean at 532 nm), the 532 nm layer-mean volume depolarization ratio, the mid-layer altitude, and the latitude. The confidence in the cloud or aerosols classification is quantified through so-called CAD scores (Liu et al., 2019).

CALIOP cloud and aerosol vertical profiles, which are available during both day and night, have been used to evaluate

cloud and aerosol discrimination from passive sensors, in particular dust detection (e.g., Zhou et al., 2020). Retrievals from passive observations in the thermal infrared spectral domain are applicable during both day and night, in contrast to multi-spectral retrievals involving channels in the near infrared or visible spectral range. Using a split-window technique (Inoue, 1985), dust or volcanic aerosols can be detected using channels centered in the atmospheric window (e.g., Ackerman, 1997; Pierangelo et al., 2004; Ashpole and Washington, 2012; Prata and Prata, 2012; Capelle et al., 2018). These aerosol layers

can be distinguished from ice clouds (Ackerman et al., 1990) through the analysis of the sign and amplitude of inter-channel brightness temperature differences (BTDs), because clouds and aerosols such as volcanic ash or dust exhibit different spectral variations of their respective complex refractive indices. This technique has proven to be useful in reducing the frequency of dense dust misclassified as cloud in the V3 CAD algorithm using on-board satellite infrared spectroradiometers (Chen et al., 2010; Naeger et al., 2013a, b).

In this paper, we take advantage of the co-located observations from the CALIPSO Imaging Infrared Radiometer (IIR) to reassess the CALIOP V4 CAD algorithm with the analysis of inter-channel BTDs in the thermal infrared spectrum. IIR includes three medium resolution channels centered at 8.65, 10.60 and 12.05 µm (Garnier et al., 2018). The IIR swath is 69-km wide, with a pixel size of 1 km, and the center of the IIR swath is by design temporally and spatially co-located with the CALIOP track, allowing detailed analyses. We develop a new IIR CAD algorithm, similar in concept to the CALIOP algorithm,

that uses two-dimensional (2-D) PDFs derived from IIR BTDs; i.e., 8.65 µm – 12.05 µm and 10.60 µm – 12.05 µm. Without additional information, the interpretation of these IIR observations is often uncertain. They vary with layer optical depth and microphysical properties, which drive the fraction of the background incoming radiance absorbed and re-emitted at layer temperature in each of the IIR channels. Here, their interpretation is informed by the vertical description of the atmospheric column seen by IIR as provided by CALIOP, namely layer altitude and inferred temperature, as well as optical depth. Multilayer

cases are not considered because passive IIR observations cannot isolate the signature from several individual layers. The BTDs are analyzed in terms of their departure from computed clear-sky conditions as an attempt to isolate the impact of faint layers. The analysis is limited to observations acquired over ocean surfaces to minimize uncertainties in clear-sky computations



(Garnier et al., 2021). Polar regions, where sea ice can arise, are excluded from the analysis for the same reason. The CAD scores in the CALIOP V4 and in the new IIR cloud–aerosol classifications are compared and cases where the CALIOP V4 algorithm could benefit from IIR observations are discussed.

Section 2 presents the IIR and CALIOP data, and the 1-D radiative transfer model. Section 3 presents the IIR 2-D BTD signatures in cloud and aerosol monolayer atmospheric columns using radiative transfer IIR simulations with a 1-D radiative transfer model and quantify the uncertainty in the observed clear-sky IIR signatures. Section 4 describes the new IIR-based CAD algorithm. Section 5 compares the IIR cloud–aerosol classifications to the results reported by the CALIOP V4 algorithm. Section 6 summarizes the main conclusions.

## 2 Data

### 2.1 IIR observations

The IIR L2 Track Data V4 provides the brightness temperatures measured at 8.65 μm, 10.60 μm, and 12.05 μm at 1-km resolution, with co-located to the lidar track. Also reported is the clear-sky brightness temperature computed using the fast-calculation radiative transfer (FASRAD) model (Dubuisson et al., 2005; Garnier et al., 2021). Here, we average those brightness temperatures over the 5-km atmospheric columns defined in the CALIOP L2 5 km Merged Layer Data Product. Then, using the CALIOP level 2 (L2) 5 km Merged Layer Data Product V4, we retain monolayer cases. Monolayer columns are defined as 5-km (15 lidar single shots) horizontal averages of CALIOP attenuated backscatter profiles in which the CALIOP layer detection algorithm identifies only a single atmospheric layer (i.e., either a cloud or an aerosol). Our analysis is based on data above midlatitudes and tropical oceans for the 12 years of the 2008–2019 time period.

### 2.2 CALIOP observations

The CALIOP L2 5 km Merged Layer Data Product V4 reports tropospheric and stratospheric cloud and aerosol detection information on a 5-km-horizontal grid. However, the amount of horizontal averaging required to detect a layer may exceed 5 km and hence the search for features is also 20 km and 80 km averaging intervals (Vaughan et al., 2009). Here, we do not retain feature layers detected with a horizontal averaging of 80 km because their optical depths are typically very small and are therefore considered as transparent for the IIR (see Appendix A). This means that a 5-km atmospheric column containing, for example, a cloud layer detected with a horizontal averaging of 5 km and another cloud layer detected with a horizontal averaging of 80 km is considered as a cloud monolayer column here. 5-km columns are composed by 15 lidar single shots (every 333 m) averaged together. When a layer is detected at 5-km-horizontal resolution in the planetary boundary layer (PBL), the CALIOP L2 processing searches within the layer to identify those especially dense regions that can be confidently detected at single-shot resolution. In those cases where clouds are detected at single-shot resolution, the original 5-km profile is reaveraged to a nominal 5-km resolution, with the data from all single shot layer detections being excluded from the newly averaged profile. This new profile is once again searched for layers whose presence would previously have been obscured by





the very strong backscatter from boundary layer clouds. This allows, for example, the detection of aerosol layers at a nominal
5-km horizontal resolution with embedded small cumulus clouds detected at single-shot resolution. 5-km columns containing
layers detected at single-shot resolution that were cleared from the original profile are not considered in this study. Indeed, the
IIR observations can be strongly influenced by these large optical depth features.

The CALIOP V4 dataset also reports top altitude $z_{\text{top}}$, the estimated optical depth $\tau$, and the CAD score calculated by the
CAD algorithm for each detected layer. Nominal CAD scores range between -100 to 100. The layer is classified as cloud when
the CAD score is positive and as aerosol when it is negative. The absolute value of the CAD score provides a confidence
level for the classification. Here, we consider *confident* layers as those for which $70 \leq |\text{CAD}_{\text{score}}| \leq 100$ following the "high"
confidence definition of Liu et al. (2009), and *ambiguous* layers as those for which $0 \leq |\text{CAD}_{\text{score}}| < 70$. There are also several
"special" CAD score values that represent classification results that are based on additional information beyond that normally
considered in the standard CAD algorithm. For example, weakly scattering features detected along the edges of ice clouds that
are initially classified as depolarizing aerosol layers are subsequently reexamined using spatial proximity analysis. As a result,
the vast majority of these layers are reclassified as "cirrus fringes" and assigned a special CAD score of 106 (Liu et al., 2019).

Figure 1 shows the occurrence of the 5-km column types derived from the CALIOP Level 2 5 km Merged Layer Data
Product V4 over ocean during the 2008–2019 period. The column types are clear sky, low ($z_{\text{top}} < 4$ km) cloud monolayer,
high ($z_{\text{top}} \geq 4$ km) cloud monolayer, cloud multilayer, low aerosol monolayer, high aerosol monolayer, aerosol multilayer, and
cloud and aerosol multilayer. First, we note that clear-sky columns are more frequent during daytime than nighttime. This is
mainly due to the fact that the lidar detection sensitivity is much lower during daytime due to background solar noise making
it more difficult to detect faint features (e.g., Thorsen et al., 2017; Toth et al., 2018). For the same reason, more monolayer
columns and fewer multilayer columns are found during daytime compared to nighttime. In the tropics, monolayer columns
represent 54 % of daytime observations and 37 % of nighttime observations. In the midlatitudes, monolayer columns represent
57 % of daytime observations and 45 % of nighttime observations. High aerosol monolayers are very rare (0.1–0.4 %). Note
that approximately half of the aerosol columns (difference between solid and transparent bars) will not be studied using the IIR
due to the presence of dense clouds detected at full resolution (333 m) which have been cleared during the CALIOP L2 data
processing. Cloud monolayer and multilayer columns (solid bars) represent 76 % of the midlatitudes CALIOP observations in
which no clouds were cleared at single-shot resolution, and 59 % in the tropics. In contrast, aerosol monolayer and multilayer
columns are much more common in the tropics (25 %) than in the midlatitudes (13 %). We see that the proportion of features
classified with an ambiguous V4 CAD score (hatched part of the solid bars) is very low (0.5–5 % for cloud monolayer; 3–12 %
for low aerosol monolayer). This means that the V4 CAD algorithm is quite confident in its ability to correctly discriminate
aerosol from cloud layers. We note that the proportions of ambiguous features ($0 \leq |\text{CAD}_{\text{score,V4}}| < 70$) in this study is lower
than those found by Liu et al. (2019) for the year 2008 ($\approx 9$ % for cloud layers; $\approx 20$ % for aerosol layers). This disparity arises
due to the different column types being examined. In particular, cloud-aerosol discrimination is more challenging in multilayer
scenes, as wavelength-dependent signal attenuation by overlying layers introduces additional uncertainties (e.g., lower signal-
to-noise ratios) when classifying lower layers. Similarly, classification uncertainties are higher when dealing layers detected
over land, layers detected at 80-km-horizontal resolution, and/or layers from which clouds detected at single-shot resolution

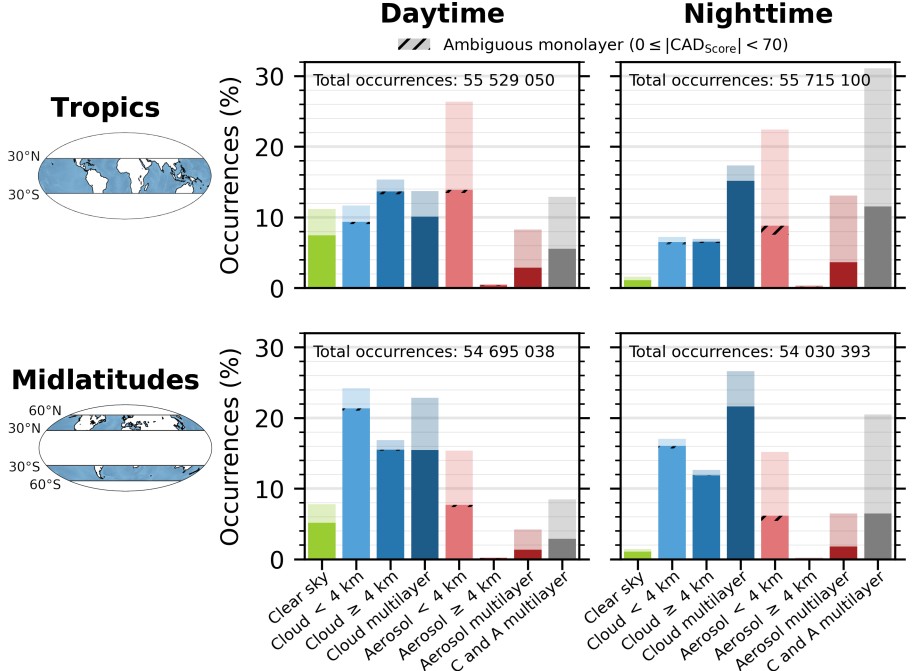

**Figure 1.** Occurrence of the 5-km atmospheric column types derived from the CALIOP L2 5 km Merged Layer Data Product V4 over ocean during the 2008–2019 period in the tropics [30° S–30° N] (top row) and the midlatitudes [30°–60°] (bottom row) during daytime (left column) and nighttime (right column). The solid bars show the occurrence frequencies of 5-km atmospheric columns in which no clouds were detected at single-shot resolution. The transparent bars show the total occurrence frequencies, i.e., including those columns in which single-shot clouds were detected. For cloud and aerosol monolayer columns (second, third, fifth, and sixth bars), the hatched region of the solid bars shows the CALIPSO V4 CAD ambiguous classification fraction. Layers detected at 80-km-horizontal averaging are not considered in the calculation of occurrence frequencies. Occurrence frequencies for multilayer columns are shown for reference but are not included in this study. Columns containing a monolayer with special CAD score value are not shown here.

have been cleared. Unfortunately, IIR will not provide any help for those column types and they are not studied here. Columns containing a monolayer with a special CAD score value are not shown in this figure. Note that monolayer columns with CAD score of 106 are more common during daytime that nighttime and represent between 0.01 % and 0.05 % of the total occurrences.

## 2.3 IIR radiative transfer simulations

In order to simulate the behavior of BTDs of ice clouds, liquid water clouds, and some types of aerosols (Sect. 3.2), simulations of the IIR signatures are performed with the Atmospheric Radiative Transfer Database for Earth and Climate Observation (ARTDECO; Dubuisson et al., 2016), a numeric tool for computing the optical properties of aerosols and clouds used for the 1-D simulation of Earth atmosphere radiances as observed with passive sensors from the UV (0.2 μm) to the far infrared (50 μm).





ARTDECO calculations also incorporate the IIR instrument functions. The 1-D radiative transfer computation is performed by the discrete ordinate method DISORT 2.1 (Stamnes et al., 1988).

The cloud optical properties used for the ice cloud simulations are computed assuming the ice cloud is composed of a generalized mixture of ice crystal habits with its size distribution defined in the microphysical model of the collection 6 data products distributed by the Moderate Resolution Imaging Spectroradiometer (MODIS) project (Baum et al., 2011). Liquid clouds are assumed to be composed of spherical water droplets with a typical stratus size distribution as defined in Stephens (1979). Optical properties of sea salt at an atmospheric relative humidity of 80 % obtained from the Optical Properties of Aerosols and Clouds (OPAC) database (Hess et al., 1998) are used for the marine aerosol layer simulations. Dust layer simulations use optical properties derived from Saharan desert dust measured in Mauritania (Di Biagio et al., 2017).

Simulations are performed for standard tropical and midlatitudes atmospheres (McClatchey, 1972) over an ocean surface with emissivities of 0.971 at 8.65 μm, 0.984 at 10.60 μm, and 0.982 at 12.05 μm, consistently with Garnier et al. (2021).

## 3 IIR signature

As introduced earlier, IIR cloud–aerosol discrimination is based on the analysis of both 8.65 μm – 12.05 μm and 10.60 μm –12.05 μm BTDs. In faint layers where CALIOP is most likely to have ambiguous cloud–aerosol discrimination (Liu et al., 2019), the layer infrared signature is better represented after subtracting the clear-sky contribution. Thus, our study is based on the *IIR signature* of the layer, that we define as the relationship between $(\mathrm{BT}_{8.65} - \mathrm{BT}_{12.05}) - (\mathrm{BT}_{8.65} - \mathrm{BT}_{12.05})^{\mathrm{CS}}$ and $(\mathrm{BT}_{10.60} - \mathrm{BT}_{12.05}) - (\mathrm{BT}_{10.60} - \mathrm{BT}_{12.05})^{\mathrm{CS}}$, where the "CS" upperscript refers to the clear-sky computations provided in 145 the IIR level 2 track product.

Note that when IIR observes a clear-sky profile or a profile containing only infrared-transparent aerosol layers, the IIR signature is zero. Therefore, if the IIR signature is non zero, the observed atmospheric profile contains a layer with a non-negligible absorption at IIR wavelengths. The accuracy of this assertion is bounded by the joint uncertainties in the measured and computed clear-sky brightness temperatures.

### 150 3.1 Observed clear-sky IIR signature uncertainty

Figure 2 shows the IIR signatures of the clear-sky atmospheric columns. Most of the observations are well centered on the origin of the figure meaning that the computed clear-sky brightness temperature correctly estimates the observed clear-sky brightness temperature. The center of the distribution is slightly shifted to the left by 0.1 K due to a small bias in the clear-sky brightness temperature more pronounced at 12.05 μm than 8.65 μm. However, the distribution is well centered in the y-axis, 155 because biases at 10.60 μm and 12.05 μm cancel each other out, consistent with Garnier et al. (2021). The red and blue ellipses represent the 95 % and 50 % confidence intervals of the 2-D gaussian PDF estimated from those observations. If a monolayer IIR signature falls into this clear-sky uncertainty region, its identification will be difficult. However, far from this region, an IIR signature can be confidently attributed to a cloud or an absorbing aerosol layer. Reliable discrimination between cloud and aerosol will be possible where their expected signature regions do not overlap.



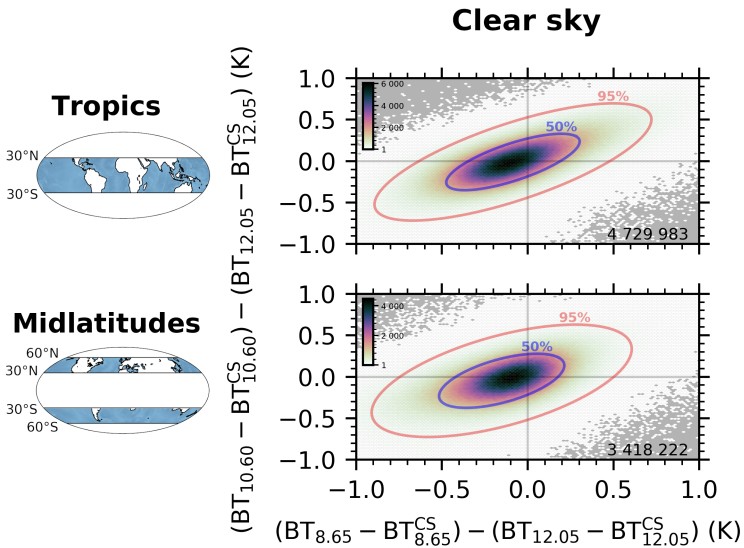

**Figure 2.** Clear-sky IIR signature (Sect. 3) observations over ocean during the 2008–2019 period shown as the difference between the 10.60 μm and 12.05 μm channels (y-axis) as a function of the difference between the 8.65 μm and 12.05 μm channels (x-axis). Top and bottom subplots show the tropics [30° S–30° N] and the midlatitudes [30°–60°] regions. The 5-km atmospheric columns are considered as clear-sky column when they do not contain any feature detected with horizontal averaging less than 80 km (so no single-shot cleared-cloud either). The colormap of each 2-D distribution linearly increases from one (white) to the maximum (black) of each subplot; no occurrence appears grey. The red and blue ellipses represent the 95 % and 50 % confidence intervals of the 2-D gaussian PDF estimated from those observations. Total number of occurrences is given on the right bottom corner of each subplot.

## 3.2 Simulated cloud and aerosol IIR signatures

In presence of a cloud or an aerosol layer, the brightness temperatures in the IIR channels depend on the layer altitude, the layer optical depth, the microphysics of the layer, the atmospheric profile, and the surface temperature and emissivity. We briefly present here how the layer parameters affect the IIR signature using the radiative transfer simulations presented in Sect. 2.3.

### 3.2.1 Clouds

Figure 3 shows how the IIR signature from an ice or liquid cloud layer varies with the cloud optical depth, the cloud top altitude, and the particle equivalent diameter in tropical and midlatitudes atmospheres. Ice cloud particle equivalent diameter of 20 μm, 40 μm, and 90 μm are used. We note that as the cloud optical depth or cloud altitude is changing, all the rest being the same, the simulated IIR signature describes arches on this representation, which is consistent with previous studies using the split-window technique (e.g.; Baum et al., 1994; Giraud et al., 1997; Dubuisson et al., 2008; Hong et al., 2010). The arches converge toward the clear-sky IIR signature (zero) as the cloud optical depth and cloud altitude decrease and toward the top-of-atmosphere black body IIR signature (red cross) as the cloud optical depth and cloud altitude increase. Indeed, if the cloud





is dense enough, its emissivity is close to one in each channel and then their brightness temperatures are the same. Moreover, if the cloud is high enough, the IIR signatures are weakly impacted by the water vapor above the cloud. As a result, the IIR signature in very dense clouds represents $-\mathrm{BTD}^{\mathrm{CS}}$ in each channel pair, i.e., $\mathrm{BT}^{\mathrm{CS}}_{12.05} - \mathrm{BT}^{\mathrm{CS}}_{10.60}$ vs $\mathrm{BT}^{\mathrm{CS}}_{12.05} - \mathrm{BT}^{\mathrm{CS}}_{8.65}$. The

red and blue dashed ellipses represent the observed 95 % and 50 % confidence intervals of IIR clear-sky observations (Fig. 2). As expected, clouds with very thin optical depths ($\tau \leq 0.1 - 0.2$) fall into this region. A cloud lying very close to the surface ($z_{\mathrm{top}} \leq 1 - 2$ km) will also fall into the clear-sky uncertainty region because the radiative contrast is very small and it is not possible to differentiate its IIR signature from the clear-sky IIR signature, unless its optical depth is large enough. Note that for liquid clouds, even optically thick and high clouds can stay in the clear-sky uncertainty region. Then, the IIR measurements of

such cloud will be of no help in assessing the CAD algorithm classification. We note that the differences in the atmospheric profiles between summer and winter at the midlatitudes affect the IIR signature but that the altitude and optical depth of the cloud are the main drivers.

### 3.2.2 Aerosols

Figure 4a shows an example of a simulation with a low-absorbing marine aerosol layer. These layers, which represent 71 % of

the aerosol monolayer columns found over ocean, provide an IIR signature inside the clear-sky signature uncertainty (red and blue dashed ellipses). The IIR observations is then of no help for the classification of such layers.

Unlike non-absorbing aerosols, the IIR signature in dust layers can be outside the clear-sky uncertainty region, as illustrated in Fig. 4b. A dust layer with $0.2 < \tau < 3$ and $z_{\mathrm{top}} \geq 4$ km has an IIR signature very different from a cloud with such optical depth and altitude (Fig. 3), in agreement with previous studies (e.g., Ackerman, 1997; Capelle et al., 2018). Using optical depth

and altitude information from CALIOP allows a more robust interpretation of the IIR signature and subsequent cloud-aerosol discrimination.

We acknowledge that these simulations do not represent an exhaustive view of the IIR signatures of clouds and aerosols since we have chosen specific compositions, microphysics, size distributions, and atmospheric profiles. The goal is to illustrate the variation of the IIR signature with altitude and optical depth and ultimately to provide insight into the possibility to discriminate

clouds from aerosols.

## 4 IIR Cloud–Aerosol Discrimination (CAD) algorithm

Following the CALIOP V4 CAD method (Liu et al., 2009, 2019), we develop a new IIR CAD algorithm. To do so, we build 2-D cloud and aerosol PDFs of the IIR signature observed in cloud and aerosol layers. These PDFs are built for several $z_{\mathrm{top}}-\tau$ ranges to account for the sensitivity of the IIR signature to these parameters (see previous section), by separating tropical and

mid-latitudes. The training data set is based on cloud and aerosol layers classified as such with confidence by the CALIOP V4 algorithm ($|\mathrm{CAD}_{\mathrm{score,V4}}| \geq 70$). This assumes that CALIOP misclassifications in confident layers are unlikely. The ability to discriminate clouds from aerosols is quantified through a so-called IIR CAD score, which characterizes the difference between the respective IIR signatures and the overlapping of the respective PDFs. We used the whole 2008–2019 time period for the



training data set. Examples of observed cloud and aerosol 2-D distributions are shown in Sect. 4.1. The distributions are shown

for the training data set, that is for confident layers, as well as for ambiguous layers, to discuss the additional value of the IIR for cloud–aerosol discrimination. The derivation of the IIR CAD scores for the measure of this additionnal value is described in Sect. 4.2, followed by presentation of CAD score masks in Sect. 4.3.

## 4.1 Cloud and aerosol PDFs of IIR signature

### 4.1.1 Clouds

Figure 5 shows the IIR signatures for ice cloud monolayer columns. Confidently classified ice clouds (Fig. 5, left) show two modes. A first mode is centered on the origin and extends into the top-right quadrant. When we compare this mode to the ice cloud simulations (Fig. 3, left), it corresponds to very thin clouds with $\tau < 0.2$. Beginning at the mode centroid and moving to the upper right along the distribution tail, we see increasing altitudes and slightly increasing optical depths (up to $\tau \approx 1$). The second mode is located in the bottom-left quadrant. Compared to ice cloud simulations (Fig. 3, left), it should

correspond to optically-thick clouds. Those two modes are connected by an arch of less numerous observations which match with the simulations. The good agreement between the observations and the simulations can be further confirmed by looking at observations with specific ranges of altitude and optical depth. Two examples are provided in Appendix B, where Figs. B1 and B2 show, respectively, high-altitude ($z_{\mathrm{top}} > 8$ km) thin $\tau < 0.2$ and thick $\tau > 3$ layers. Note that, in agreement with the simulations, high dense clouds do not exhibit the same IIR signature in the tropics and the midlatitudes. Their 10.60 μm–

12.05 μm component (y-axis) of the IIR signature is closer to zero in the midlatitudes, whereas their 8.65 μm–12.05 μm component (x-axis) is negative in the tropics but positive in the midlatitudes. Because the ice–water phase algorithm is applied to cloud layers with V4 CAD score larger than 20 (Avery et al., 2020), the ambiguous ice clouds (Fig. 5, right) have a CAD score larger than 20 (and smaller than 70). Many of these ambiguously classified ice clouds are close to the origin and within the clear-sky uncertainty region (red and blue dashed ellipses). As a result, these layers cannot be confidently confirmed with IIR

observations. However, the ambiguous layers outside the clear-sky uncertainty region could potentially be confirmed thanks to the IIR measurements. The faint mode found in the bottom-left quadrant in the tropics plot corresponds mainly to thick clouds in the upper troposphere (Fig. B2). They get ambiguous CALIOP V4 CAD scores because their layer-mean attenuated backscatters at 532 nm ($\approx 0.02$ km$^{-1}$ sr$^{-1}$) are larger than usually found in cirrus clouds (Liu et al., 2019) and their layer-mean 532 nm volume depolarization ratios are quite low (0.2–0.35).

Figure 6 shows the IIR signatures obtained for liquid clouds. There is a good agreement with simulations (Fig. 3, right) with many thick clouds located in the bottom-left quadrant and a very faint tail in the top-right quadrant corresponding to intermediate optical depths ($0.2 < \tau < 7$) and intermediate altitudes ($4 < z_{\mathrm{top}} < 8$ km). Separation of liquid cloud observations in $z_{\mathrm{top}}$–$\tau$ classes further confirms this good agreement (see Figs. C1 and C2). A discernible fraction of tropical ambiguous liquid cloud are detected by the IIR outside the clear-sky uncertainty region and could potentially be confirmed thanks to IIR

observations.





### 4.1.2 Aerosols

Figure 7 shows the results for CALIOP-classified clean marine layers observed by IIR. 99 % of those layers are found with a layer top altitude less than 4 km and optical depth less than 0.2. The red and blue solid line ellipses represent the observed 95 % and 50 % confidence intervals of the IIR observations for the confident marine layers and are similar to the dashed line
ellipses corresponding to those of clear-sky regions shown in Fig. 2, thereby confirming that these non-absorbing aerosols have no IIR signature.

    Figure 8 shows the results for dust layers identified by CALIOP with $\tau < 0.2$ and $4\,\mathrm{km} < z_{\mathrm{top}} < 8\,\mathrm{km}$. Those layers absorb and reemit the infrared radiation. Confidently classified dust IIR signatures are therefore shifted from the clear-sky signatures to the bottom-left quadrant, in good agreement with the dust simulations (Fig. 4b). For the ambiguous tropical layers, the IIR
signature is outside the clear-sky uncertainty region for only a very small number of layers. This number slightly increases in the midlatitudes.

    Simulations of dust IIR signatures in Fig. 4b suggest that a high ($z_{\mathrm{top}} > 4$ km) and dense ($\tau > 1$) dust layer could provide a large negative IIR signature. Observed dense dust case studies over land desert (Chen et al., 2010; Naeger et al., 2013a, b) also confirm the results found with the radiative transfer simulations. Then, we propose to check how far from the origin a negative
IIR signature of a dust layer can be in the 2-D BTD representation. In order to do that, we select an oceanic region close to dust source defined as 10° N–38° N and 25° W–65° E that we call the "dust belt". Figure 9 shows dust, ice cloud, and liquid cloud IIR BTD signatures for layers with $4\,\mathrm{km} < z_{\mathrm{top}} < 8\,\mathrm{km}$ and $0.2 < \tau < 3$ (dense dust layers at higher altitude and/or larger optical depth are virtually impossible). We note that dust layers can indeed provide quite large negative IIR BTD signatures. We note that these dust IIR signatures do not overlap with those of ice and liquid clouds for such $z_{\mathrm{top}}$–$\tau$ ranges making the
discrimination between dense dust layers and cloud layers possible using IIR observations.

## 4.2 IIR CAD score

The IIR signature 2-D PDFs derived from confident observations of a specific $z_{\mathrm{top}}$–$\tau$ class, as illustrated by their 95 % and 50 % confidence intervals (solid lines) in Figs. B1, B2, C1, C2, 7, and 8, are used to derive an IIR CAD score.

    The IIR signature PDFs are derived for several $z_{\mathrm{top}}$–$\tau$ ranges in order to keep the PDF widths as small as possible as we
saw in previous section that the IIR signatures are mainly dependent on layer altitude and optical depth. These narrower PDFs increase the likelihood that the PDFs of individual cloud and aerosol classes are well separated. Probabilities are then computed on a $z_{\mathrm{top}}$–$\tau$ grid, with $z_{\mathrm{top}}$ boundaries from 0–4 km, 4–8 km, and above 8 km and $\tau$ spanning ranges from 0–0.2, 0.2–0.6, 0.6–1.5, 1.5–3, and above 3. The PDFs $p_i$, where $i$ represents all cloud types (liquid, ice, oriented crystals) and all aerosol types (dust, smoke, marine, ...) of the troposphere and stratosphere of the CALIPSO V4 classification, are then defined for each
$z_{\mathrm{top}}$–$\tau$ grid cell. PDFs characterizing specific layer types are derived whenever there are at least 500 confident occurrences of



layer type $i$ in a $z_{\mathrm{top}}$–$\tau$ grid cell. Then, we derive the IIR CAD score according to:

$$
\mathrm{CAD}_{\mathrm{score,IIR}} = \begin{cases} \min(\mathrm{CAD}_{\mathrm{noCS}}, \mathrm{CAD}_{\mathrm{Cloud/CS}}) \\ \quad \text{where } \mathrm{CAD}_{\mathrm{noCS}} \geq 0 \\ \min(\mathrm{CAD}_{\mathrm{noCS}}, \mathrm{CAD}_{\mathrm{Aerosol/CS}}) \\ \quad \text{where } \mathrm{CAD}_{\mathrm{noCS}} < 0 \end{cases} \tag{1}
$$

with

$$
\mathrm{CAD}_{\mathrm{noCS}} = 100 \frac{(P_C + P_{\mathrm{bkg}}) - (P_A + P_{\mathrm{bkg}})}{(P_C + P_{\mathrm{bkg}}) + (P_A + P_{\mathrm{bkg}})}(1 + 2P_{\mathrm{bkg}}) \tag{2}
$$

representing the CAD score if there were no clear-sky atmospheric columns (or no uncertainty in the computed clear-sky brightness temperatures),

$$
\mathrm{CAD}_{\mathrm{Cloud/CS}} = 100 \frac{(P_C + P_{\mathrm{bkg}}) - (kP_{\mathrm{CS}} + P_{\mathrm{bkg}})}{(P_C + P_{\mathrm{bkg}}) + (kP_{\mathrm{CS}} + P_{\mathrm{bkg}})}(1 + 2P_{\mathrm{bkg}}) \tag{3}
$$

representing the CAD score of clouds if there were only cloud and clear-sky atmospheric columns, and

$$
\mathrm{CAD}_{\mathrm{Aerosol/CS}} = 100 \frac{(kP_{\mathrm{CS}} + P_{\mathrm{bkg}}) - (P_A + P_{\mathrm{bkg}})}{(kP_{\mathrm{CS}} + P_{\mathrm{bkg}}) + (P_A + P_{\mathrm{bkg}})}(1 + 2P_{\mathrm{bkg}}), \tag{4}
$$

representing the CAD score of aerosols if there were only aerosols and clear-sky atmospheric columns. In those equations, $P_{\mathrm{CS}}$ corresponds to the clear-sky PDF weighted by a coefficient $k = 2$ in order to decrease the absolute value of IIR CAD score in the clear-sky uncertainty region. The cloud and aerosol PDFs are given by

$$
P_C(X_1, X_2, ..., X_m) = \max_{i \in \text{cloud types}} p_i(X_1, X_2, ..., X_m) \tag{5}
$$

and

$$
P_A(X_1, X_2, ..., X_m) = \max_{i \in \text{aerosol types}} p_i(X_1, X_2, ..., X_m) \tag{6}
$$

where $p_i$ are the multidimensional PDFs for cloud and aerosol types as a function of attributes $X_1$, $X_2$, ..., $X_m$. A background PDF $P_{\mathrm{bkg}} = 0.05$ is added to the equations in order to avoid unreasonably large CAD values in the regions that both cloud and aerosol would not present by nature. The CAD score equations are then renormalized by multiplying them by $(1 + 2P_{\mathrm{bkg}})$.

Attributes $X_1$, $X_2$, ..., $X_m$ are both components of the IIR signature (i.e., 8.65 μm–12.05 μm and 10.60 μm–12.05 μm),
the top altitude $z_{\mathrm{top}}$ and optical depth $\tau$ of the monolayer inferred from lidar observables, and the latitude to determine the region (tropics or midlatitudes). Unlike the CAD score derived solely from lidar observations (Liu et al., 2009), the CAD score from IIR observations does not account a priori for the relative occurrence frequencies of different layer types within a $z_{\mathrm{top}}$–$\tau$ grid cell. We then consider that the probability of occurrence of an $i$th type of cloud or aerosol with a given IIR signature is independent of the probability of occurrence of another type. The maximum of PDFs for the different cloud or aerosol types
are then considered to compute the CAD score instead of merging the different PDFs when comparing $P_C$ and $P_A$.



### 4.3 IIR CAD score masks

Figures 10 and 11 show the IIR CAD scores derived from CALIPSO observations for the tropics and the midlatitudes respectively. Aerosol classifications are shown in red and cloud classifications are shown in blue. Each pattern is due to a specific layer type, some of them being annotated in Fig. 10. Color intensity varies according to classification confidence, with fainter
colors representing lower confidence, which decreases with the distance to the PDF centers. The yellow lines represent the $|\mathrm{CAD}_{\mathrm{score,IIR}}| = 70$ isocontours, separating ambiguous and confident IIR classifications.

As expected according to its definition, the IIR CAD score is very close to 0 in the clear-sky uncertainty region. In the tropics, we note that for low altitude layers ($z_{\mathrm{top}} < 4$ km; left column), the intensity of the colors is quite faint (unless the optical depth is very large), meaning the IIR CAD score is almost never confident. Indeed, the discrimination between cloud
and aerosol is difficult for low layers due to the lack of contrast in their brightness temperatures compared to the surface.

Mid and high-altitude layers ($z_{\mathrm{top}} > 4$ km; middle and right columns) are easier to discriminate both in the tropics and the midlatitudes. Layers with an IIR signature falling in the red regions are almost always classified as aerosol without confidence due to overlap with cloud PDFs in the IIR signature regions where we found them. The only exception arises in the midlatitudes for layers at mid-altitudes and with optical depths between 0.2 and 1.5.

## 5 Results

### 5.1 IIR CAD score vs V4 CAD score

Figure 12 compares CAD score of the V4 algorithm with those derived from IIR BTD observations for all monolayer columns observed by CALIPSO during the 2008–2019 period. Transparent white lines show the limit between confident and ambiguous cases ($|\mathrm{CAD}_{\mathrm{score}}| = 70$) and between cloud and aerosol CALIOP V4 classification ($|\mathrm{CAD}_{\mathrm{score,V4}}| = 0$). For IIR classification,
we consider CAD scores very close to zero ($|\mathrm{CAD}_{\mathrm{score,IIR}}| < 10$) as undefined classification. Therefore, cloud and aerosol ambiguous layers have CAD scores of $10 < |\mathrm{CAD}_{\mathrm{score,IIR}}| < 70$. Tables below the plots summarize the fractions in these CAD score classes. We first note that the CALIOP V4 algorithm is generally very confident in its ability to discriminate cloud and aerosol, as seen by the many values very close to 100 and -100, consistent with Liu et al. (2019). When the V4 detection is ambiguous ($|\mathrm{CAD}_{\mathrm{score,V4}}| < 70$), the CAD score is mainly very close to 0 (peak around 0). A large majority of the CALIOP
V4 confidently classified clouds are also confidently classified as clouds (ambiguous or confident) by the IIR (91 % in the tropics, 86 % in the midlatitudes). Very few V4 confidently classified clouds are classified as ambiguous aerosols by the IIR CAD algorithm ($\approx 0.15$ %) and virtually none as confident aerosols. Some of the V4 ambiguous clouds can be confirmed thanks to IIR observations (28 % in the tropics and 14 % in the midlatitudes) as they received a confident IIR CAD score. The V4 confident aerosols are mainly undefined by the IIR CAD algorithm (84 % in the tropics and 87 % in the midlatitudes).
A few occurrences of V4 aerosols with an IIR confident cloud classification are found in both the tropics (0.11 %) and the midlatitudes (0.22 %).



We now compare the IIR and V4 CAD scores for each cloud and aerosol type. Figure 13 shows the confusion matrices between confident and ambiguous cloud and aerosol IIR and V4 CAD scores for each cloud and aerosol type observed in the tropics. The ambiguous IIR CAD score gathers the ambiguous cloud, aerosol, and undefined layers. The first and second

columns show the result for the aerosol types and the last column for the cloud types. Ice clouds, liquid clouds, and oriented ice crystal clouds of the V4 classification show a very good agreement with IIR classification since they are mainly classified as confident clouds by IIR ($> 75$ %) and never classified as confident aerosols. There are more ambiguous IIR CAD score for liquid clouds because they are mainly found at low altitudes where IIR signatures of aerosol and cloud are more difficult to untangle (see previous sections). A few of those cloud types are classified as ambiguous clouds by the V4 algorithm (0.4

to 1.7 %) while a large fraction of them can be confirmed as being clouds by the IIR observations: 48 % of ice clouds, 50 % of liquid clouds, and 89 % of oriented ice crystals. It represents a small fraction of the total observations of each cloud type because ambiguous clouds are only 1.5 % of the clouds with a known phase type. Unknown phase cloud observations are mainly ambiguous in the V4 classification (86.1 %). The optical depths of these layers are very low, and this is the reason why they are mainly classified as ambiguous by the IIR CAD score (81.1 %). We note that 60 % of the 13.9 % V4 confident

unknown phase clouds are classified as confident clouds by the IIR algorithm. 12.3 % of the ambiguous unknown phase clouds can be confirmed as being clouds by the IIR observations. They are mainly low ($z_{\text{top}} < 4$ km) thick ($\tau > 3$) layers and their IIR signatures are consistent with liquid cloud (not shown). The IIR observations are then also useful to help in the cloud phase discrimination as shown in Garnier et al. (2021).

For the aerosol observations, we are mainly interested in ambiguous cases that could be misclassified by the CALIOP CAD

algorithm but subsequently reclassified correctly as confident clouds by the IIR CAD score. A small fraction of ambiguous dust (10.1 %), polluted dust (25 %), and tropospheric elevated smoke (6.4 %) seem to be misclassified clouds. Even a small fraction of V4 confident aerosols seems to be misclassified clouds: 0.3 % for dust, 1 % for polluted dust, and 0.4 % for tropospheric elevated smoke.

Figure 14 shows the confusion matrices obtained in the midlatitudes. Some of the clouds that are ambiguously classified by

the V4 CAD can be confirmed using the IIR observations in this region: 33 % for ice clouds, 25 % for liquid clouds, 64 % for oriented ice crystals and 5 % for unknown phase clouds. A small fraction of the aerosol layers identified as ambiguous by the V4 CAD would be reclassified as clouds by the IIR observations in this region: 10 % for dust, 6.8 % for polluted dust, and 4.8 % for tropospheric elevated smoke. Some V4 confident aerosols seems also to be misclassified clouds: 0.5 % for dust, 0.2 % for dusty marine, 0.7 % for polluted dust, 0.1 % for clean marine, 0.1 % for polluted continental smoke, and 0.6 %

for tropospheric elevated smoke. A very few confirmations of ambiguous polluted dust (0.6 %) and ambiguous tropospheric elevated smoke (1.6 %) layers occur in the midlatitudes.

Since IIR confident cloud CAD score is used either to confirm V4 ambiguous clouds or to reclassify V4 ambiguous dust or smoke layers, the IIR signature dependency with $z_{\text{top}}$ and $\tau$ must be analyzed. Figure 15 shows the results obtained for V4 ambiguously classified ($0 \leq |\text{CAD}_{\text{score,V4}}| < 70$) ice clouds in the tropics. The blue line contour shows the regions where the

layer get a confident IIR CAD score ($|\text{CAD}_{\text{score,IIR}}| \geq 70$). We note that many of those V4 ambiguous cloud classification are





confirmed by the IIR CAD score, especially at altitudes above 8 km where almost 100 000 observations can be confirmed (blue values in the bottom-right corner), where more than half of them has $\tau > 3$.

Figure 16 shows the characteristics of the V4 ambiguous dust layers reclassified as clouds in the tropics. The largest number (a few thousands) of ambiguous dust layers detected as confident cloud layers by the IIR is found at high altitude for $\tau < 0.6$. For $\tau > 1.5$ most of the ambiguous dust layers are detected as clouds by the IIR.

## 5.2 Cirrus fringes

Figure 17 shows high clouds with a V4 special CAD score of 106. They represent layers initially classified as aerosols but subsequently reclassified as clouds by the V4 "cirrus fringe amelioration" algorithm (Liu et al., 2019). These layers are mainly very thin (95 % with $\tau < 0.2$) and at high altitude (97 % with $z_{\text{top}} > 8$ km in the tropics and 22 % with $4$ km $< z_{\text{top}} < 8$ km and 78 % with $z_{\text{top}} > 8$ km in the midlatitudes). The IIR signature of most of them falls in the clear-sky uncertainty region, and thus IIR cannot provide a confirmation that they are clouds on an individual basis. However, we note that the centroid of the distribution is discernibly shifted to the right-top from the clear-sky uncertainty region, which is consistent with high thin cirrus (Fig. B1). In total, 11 % of the cirrus fringes are classified as confident clouds by the IIR algorithm. These IIR observations seem to provide independent global evidence that the feature type reclassifications made by the CALIOP V4 "cirrus fringe amelioration" algorithm are indeed correct.

## 5.3 Example of cloud misclassified as dust by the V4

Figure 18 shows an example of a V4 ambiguously classified ($\text{CAD}_{\text{score,V4}} = -36$) dust layer confidently reclassified as a cloud ($\text{CAD}_{\text{score,IIR}} = 99$) using the IIR observations. The layer is located at about 12 km altitude in the 5-km atmospheric column indicated by the dash line and located at 30.6° S. Its optical depth is estimated at 0.63 in the CALIPSO product. It then falls in the $z_{\text{top}} > 8$ km and $0.6 < \tau < 1.5$ grid cell of the midlatitudes region (Fig. 11). Its IIR signature is quite large: $(\text{BT}_{8.65} - \text{BT}_{12.05}) - (\text{BT}_{8.65} - \text{BT}_{12.05})^{\text{CS}} = 4.65$ K and $(\text{BT}_{10.60} - \text{BT}_{12.05}) - (\text{BT}_{10.60} - \text{BT}_{12.05})^{\text{CS}} = 0.93$ K. This corresponds to an IIR CAD score of 99. Note that the marine aerosol layer detected below in the boundary layer (Fig. 18d) is detected with 80-km-horizontal averaging and are then not seen by IIR.

The spatial structure and magnitudes of the attenuated backscatter signal (Fig. 18a) is reasonably consistent with what we would expect from a cirrus cloud. In fact, most of the feature is correctly classified as cloud by the V4 CAD algorithm (Fig. 18b). Furthermore, Navy Aerosol Analysis and Predictions System (NAAPS) simulations (Lynch et al., 2016) show there is no dust transport from Australia for this day. Then, we are very confident the section layer classified as aerosol by the V4 algorithm (Fig. 18c) can be reclassified as cloud layer.

This case study confirms that for monolayer columns IIR observations can be useful to correct misclassified aerosol layer by the V4 CAD algorithm.



## 6    Conclusions

This paper describes how the IIR brightness temperature observations can be used to discriminate aerosol from cloud monolayers. 1-D radiative transfer simulations have been performed first to gain insight into how the IIR BTD signature evolves with increasing optical depth and altitude of clouds and aerosols. Those simulations have been compared to monolayer observations
confidently classified as cloud and can explain the IIR BTD signature behavior in $z_{\mathrm{top}}-\tau$ grid cells. An IIR CAD score based on the relative magnitude of IIR BTD signature of V4 confident observations of clouds and aerosols in each grid cell is then proposed.

Comparison between V4 and IIR CAD scores of V4 clouds shows a very good agreement with 88 % of V4 confident clouds classified as clouds by IIR. A small fraction ($\approx 1.5$ %) of the ice, liquid, and oriented ice crystal clouds are ambiguous in the
V4 classification. About 28 % and 14 % of these cases would be confidently classified as clouds using IIR observations in the tropics and the midlatitudes respectively. Unknown phase cloud layers are mainly ambiguous in the V4 and 12.3 % and 5.4 % of them, in the tropics and the midlatitudes respectively, are confirmed by the IIR observations. They mainly correspond to liquid clouds according to their IIR signature.

Comparison between V4 and IIR CAD scores of V4 aerosols show less good agreement with most of V4 aerosol layers
classified as ambiguous undefined layers by IIR. This is due to a larger uncertainty in the discrimination of aerosols and clouds by IIR at low altitude and for optically thin layers. 10 %, 16 %, and 6 % of the ambiguous V4 dust, polluted dust, and tropospheric elevated smoke respectively are found to be misclassified cloud layers by the IIR measurements. A specific analysis of a case study of the misclassified dust layers observed by CALIOP have shown to be consistent with the classification as a confident cloud proposed by the IIR CAD score. Confirmation of aerosol layers or reclassification of cloud layers in aerosol
layers virtually never occur.

Layers classified as "cirrus fringes" by the V4 CAD algorithm are optically very thin layers and have IIR signatures very similar to clear sky. However, compare to clear-sky IIR signature PDF, there is an offset of the "cirrus fringes" IIR signature PDF toward the cirrus IIR signatures and 11 % of the cirrus fringes even have a confident IIR cloud CAD score, suggesting the "cirrus fringes amelioration" algorithm in the CALIOP V4 CAD processing make correct cloud classifications.
This study mainly confirms the overall very good working of the V4 CAD algorithm using the independent observations of the IIR radiometer. For a few cases the IIR can even improve the CALIOP based CAD by confirming part of the remaining ambiguous cloud layers and correct some aerosol layer misclassifications often embedded in larger cloud structures. IIR can also help with the cloud–aerosol classification where lidar signal is very noisy or when low laser energy shots occur. The preliminary version of a CAD score based on the IIR observations proposed in this paper could serve as a first step for the
design of a future CAD making use of both the lidar and the IIR observations.

*Data availability.* The CALIPSO Lidar Level 2 5 km Merged Layer Data Version 4-20 are available at: http://doi.org/10.5067/CALIOP/
CALIPSO/LID_L2_05KMMLAY-STANDARD-V4-20, last access: 28 July 2021. The CALIPSO Infrared Imaging Radiometer Level 2





## Appendix A: IIR signature of layers detected with 80-km horizontal averaging

Here we propose to check if the radiative impact of 80-km horizontal averaging layers can affect the IIR measurements of a layer found below. A layer detected at 80-km horizontal averaging is optically very thin. In order to impact the IIR measurement, its emission temperature needs to be very different from that of the feature found below. In other words, a 80-km horizontal averaging layer needs to be very high (then cold) to get a chance to affect the measurements. Then it also needs not to be optically too thin, otherwise it would have no effect.

We retain all uppermost layers (first layer from top detected in an atmospheric column) detected at 80-km horizontal averaging with a top layer altitude above 8 km for the 2008 year. Figure A1 shows the cumulative distribution function (CDF) of their integrated attenuated backscatter at 532 nm $\gamma'_{532}$. We note that 99 % of these layers have $\gamma'_{532} \leq 0.00207$. We will use this value to consider the worst possible case.

We now derive the optical depth $\tau$ of an ice cloud at 8, 12, and 16 km and a liquid cloud at 8 km with $\gamma'_{532} = 0.00207$. In a tropical atmosphere (McClatchey, 1972), the temperature at 8, 12, and 16 km is -23, -49, and -76 °C respectively. According to (Young et al., 2018), for an ice cloud, the multiple scattering factor $\eta$ for those atmospheric temperatures is 0.48, 0.57, and 0.73 respectively, and the lidar ratio $S$ is 35, 33, and 23 respectively. For a liquid cloud at 8 km (-23 °C), we have $\eta = 0.44$ and $S = 18$.

From those values and according to Platt (1973) equation:

$$\tau = -\frac{1}{2\eta}\ln(1 - 2\eta S\gamma'_{532}), \tag{A1}$$

we obtain for the ice cloud $\tau = 0.075$ at 8 km, $\tau = 0.071$ at 12 km, and $\tau = 0.049$ at 16 km. For a liquid cloud at 8 km $\tau = 0.038$.

Finally, we simulate the IIR signature (see Sect. 3) of these worst case scenarios in a 1-D radiative transfer code. Figure A2 shows the results. We note that even for these worst case scenarios, the IIR signature stays in the clear-sky uncertainty region (see Sect. 3.1). Then, we can confidently remove the layers at 80-km horizontal averaging since their effect to the IIR measurements is negligible.

Note that a layer found at 80-km horizontal averaging below another layer could have been detected with higher resolution if the top layer was not present. It means that such layer could have an optical depth larger than those found here. However, if such a layer is detected at 80-km horizontal averaging and not at a higher resolution, it means that the layer above is quite thick (in addition to be colder) and then clearly dominates the IIR signature.

## Appendix B: Examples of ice cloud in specific $z_{\text{top}}$–$\tau$ grid

Figs. B1 and B2.





## Appendix C: Examples of liquid cloud in specific $z_{\mathrm{top}}$–$\tau$ grid

Figs. C1 and C2.

*Author contributions.* GA and JP proposed the study. TVG and GA led the design of the method. TVG performs the results and wrote the manuscript. AG provides the knowledge of the IIR measurements and products. LL helps for the implementation of radiative transfer numerical simulations using ARTDECO. All the authors contributed to discussion and feedback essential to the study.

*Competing interests.* The authors declare that they have no conflict of interest.

*Acknowledgements.* This work has been financially supported by the French government space agency CNES under the grant BC60466 and the NASA Langley Research Center under the grant NNL16AA05C. The AERIS infrastructure is gratefully acknowledged for maintenance of the CALIPSO data base and computer resources for data processing.



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

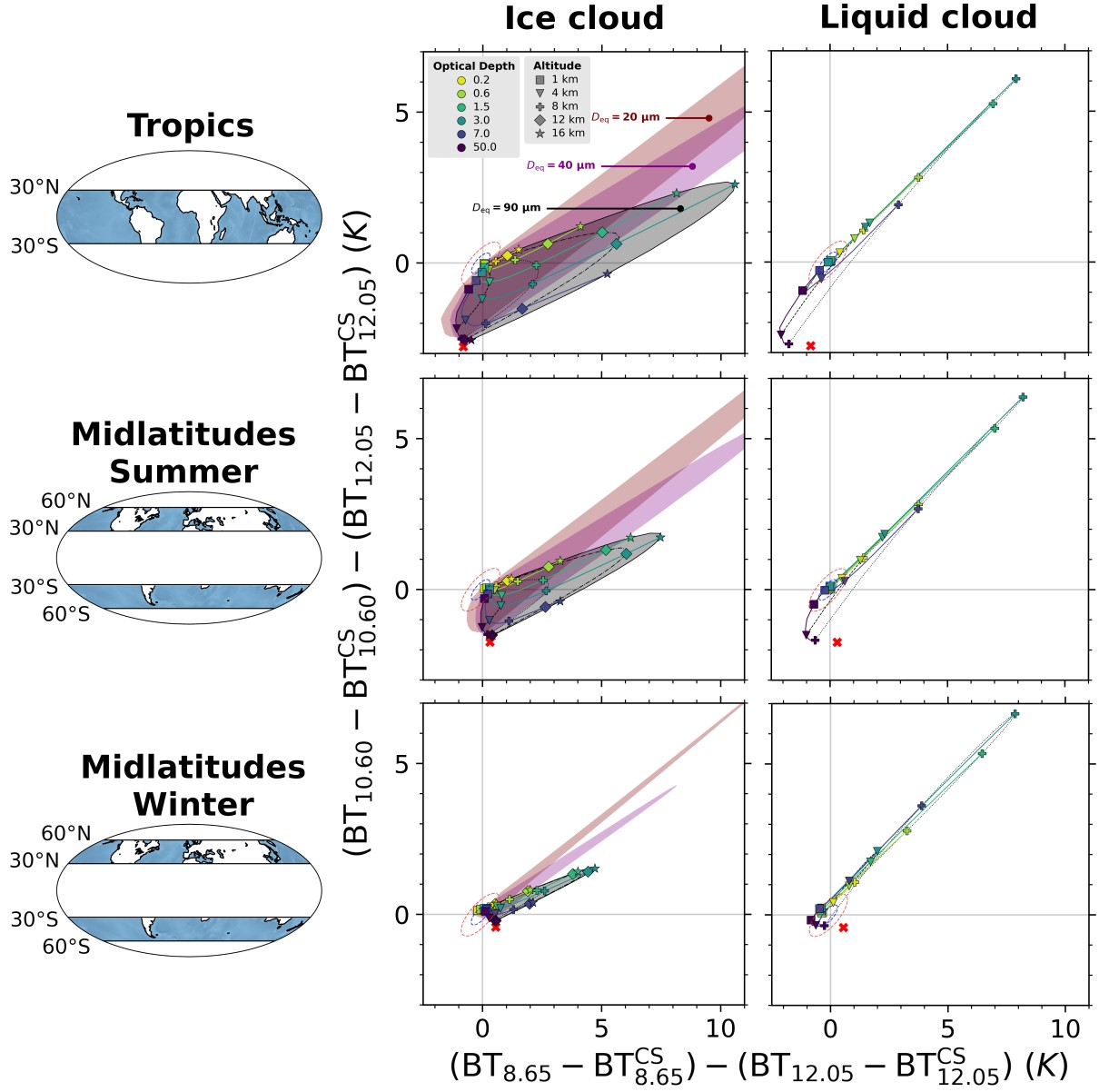

**Figure 3.** Radiative transfer simulations of the evolution of IIR signature (Sect. 3) with altitude and optical depth of an ice cloud (left column) and a liquid cloud (right column) with 1-km-geometrical thickness in a tropical atmosphere (top row), a midlatitudes summer atmosphere (middle row), and a midlatitudes winter atmosphere (bottom row). The ice cloud particle size distribution is a general habit mixed particle distribution with an equivalent diameter $D_{eq} = 90$ μm. The liquid cloud particle size distribution is typical of a stratus cloud. Cloud visible (550 nm) optical depth (color) and top cloud altitude (marker shape) variations draw arches. Locations of those arches for ice cloud with $D_{eq} = 20$ μm and $D_{eq} = 40$ μm are shown in maroon and purple. The red cross shows the IIR signature of a top-of-atmosphere black body: $BT_{12.05}^{CS} - BT_{10.60}^{CS}$ vs $BT_{12.05}^{CS} - BT_{8.65}^{CS}$. The red and blue dashed ellipses represent the observed 95 % and 50 % confidence intervals of IIR signature clear-sky observations (see Fig. 2).





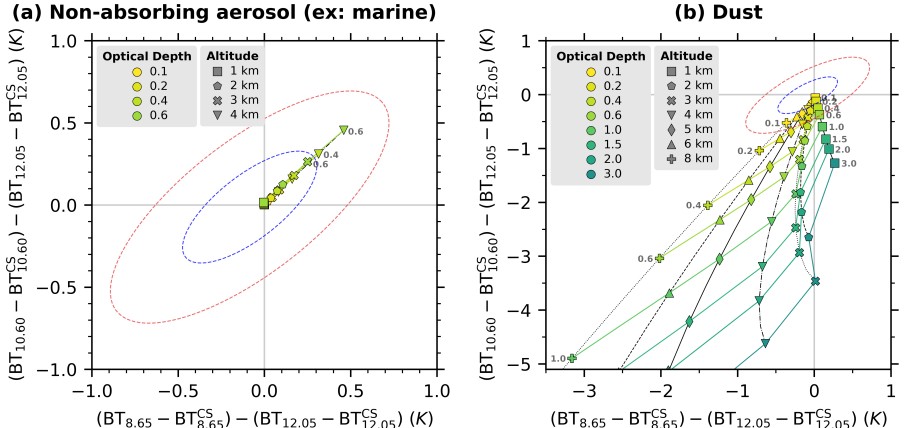

**Figure 4.** Radiative transfer simulations of the evolution of IIR signature (Sect. 3) with altitude and optical depth of (a) a non-absorbing aerosol layer and (b) an absorbing aerosol layer. Geometrical thickness of the layer is 1 km. Optical properties of sea salt with relative humidity of 80 % are used for the marine aerosol simulation. Optical properties of dust of Sahara desert in Mauritania are used for the dust simulation. The atmospheric profiles used for the simulations are those for a tropical atmosphere. The red and blue dashed ellipses represent the observed 95 % and 50 % confidence intervals of IIR clear-sky observations (see Fig. 2).

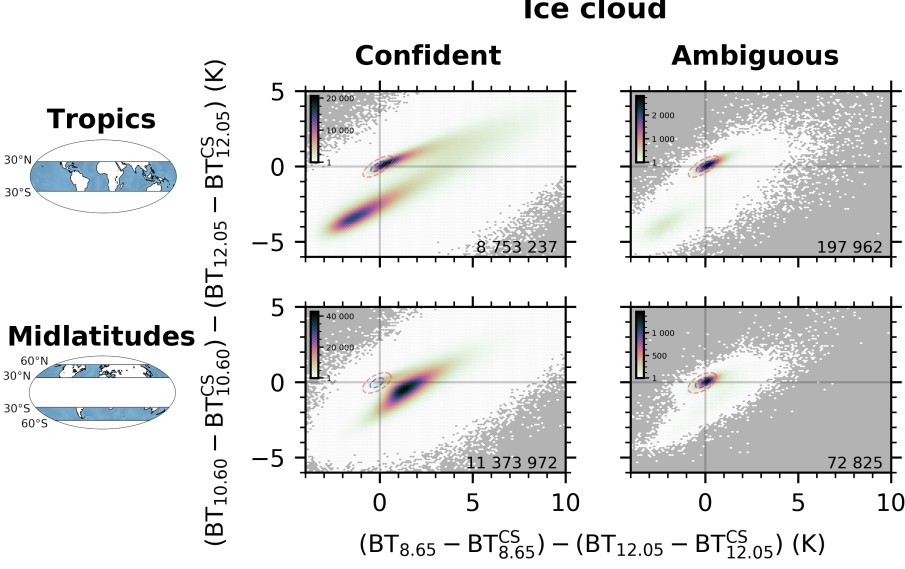

**Figure 5.** As Fig. 2 for ice cloud monolayer observations considered as confident ($70 \leq |\mathrm{CAD}_{\mathrm{score,V4}}| \leq 100$) (left column) and ambiguous ($0 \leq |\mathrm{CAD}_{\mathrm{score,V4}}| < 70$) (right column) by the CALIPSO V4 algorithm. The red and blue dashed ellipses represent the observed 95 % and 50 % confidence intervals of IIR clear-sky observations (see Fig. 2).

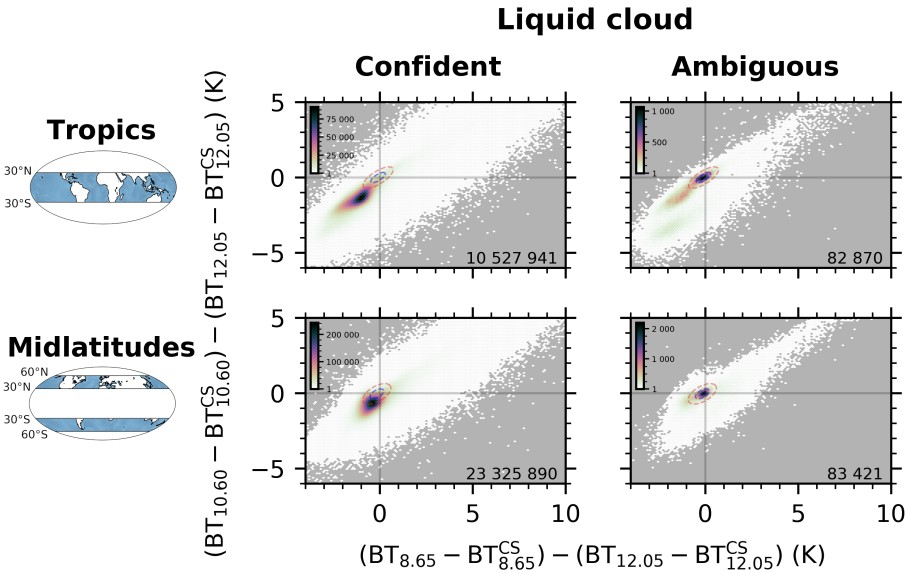

**Figure 6.** As Fig. 5 for liquid cloud monolayer observations.

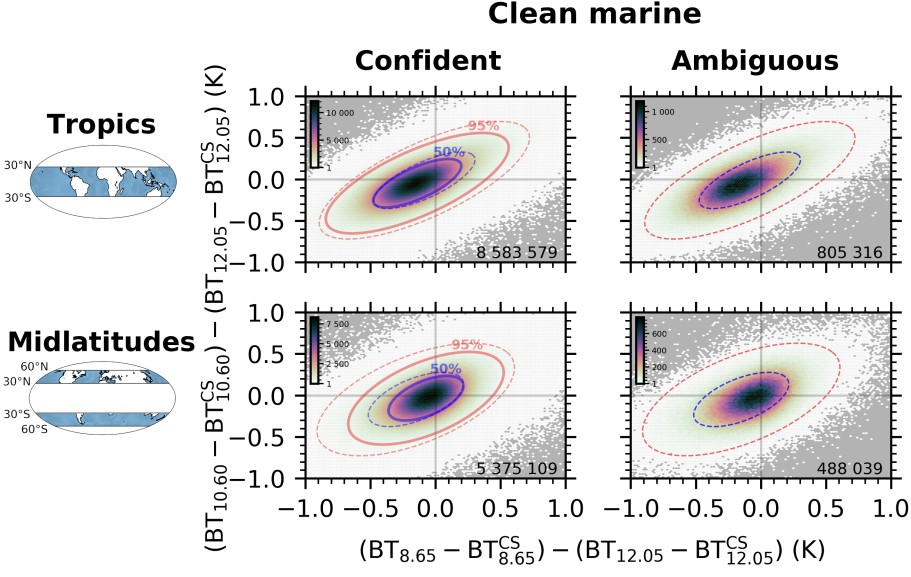

**Figure 7.** As Fig. 5 for clean marine monolayer observations with $\tau < 0.2$ and $z_{\text{top}} < 4$ km (99 % of all clean marine observations). The red and blue solid ellipses represent the 95 % and 50 % confidence intervals of the 2-D gaussian PDF estimated from the confident observations of this specific $z_{\text{top}}$–$\tau$ grid and used in the IIR CAD score definition (Eq. (1)).





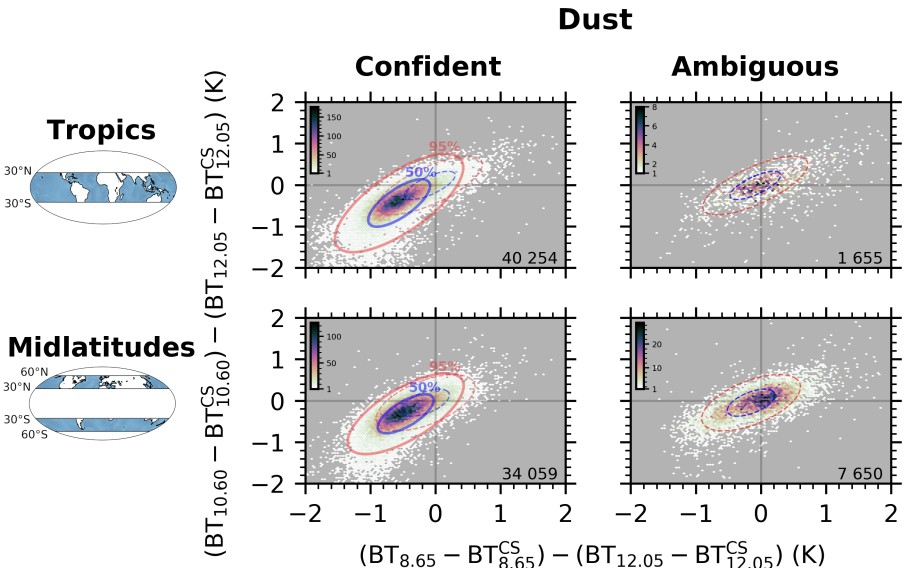

**Figure 8.** As Fig. 7 for dust monolayer observations with $\tau < 0.2$ and $4\,\text{km} < z_{\text{top}} < 8\,\text{km}$.

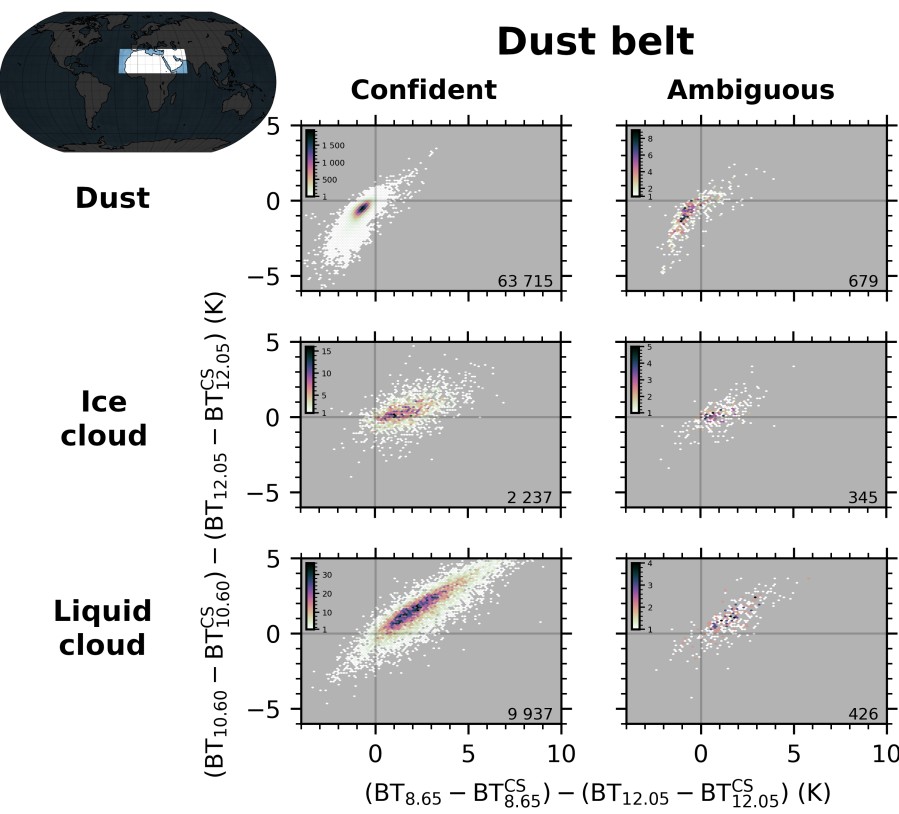

**Figure 9.** As Fig. 2 for dust, ice cloud, and liquid cloud detected in the dust belt region (10° N–38° N and 25° W–65° E) with $4 < z_{\text{top}} < 8$ km and $0.2 < \tau < 3$.

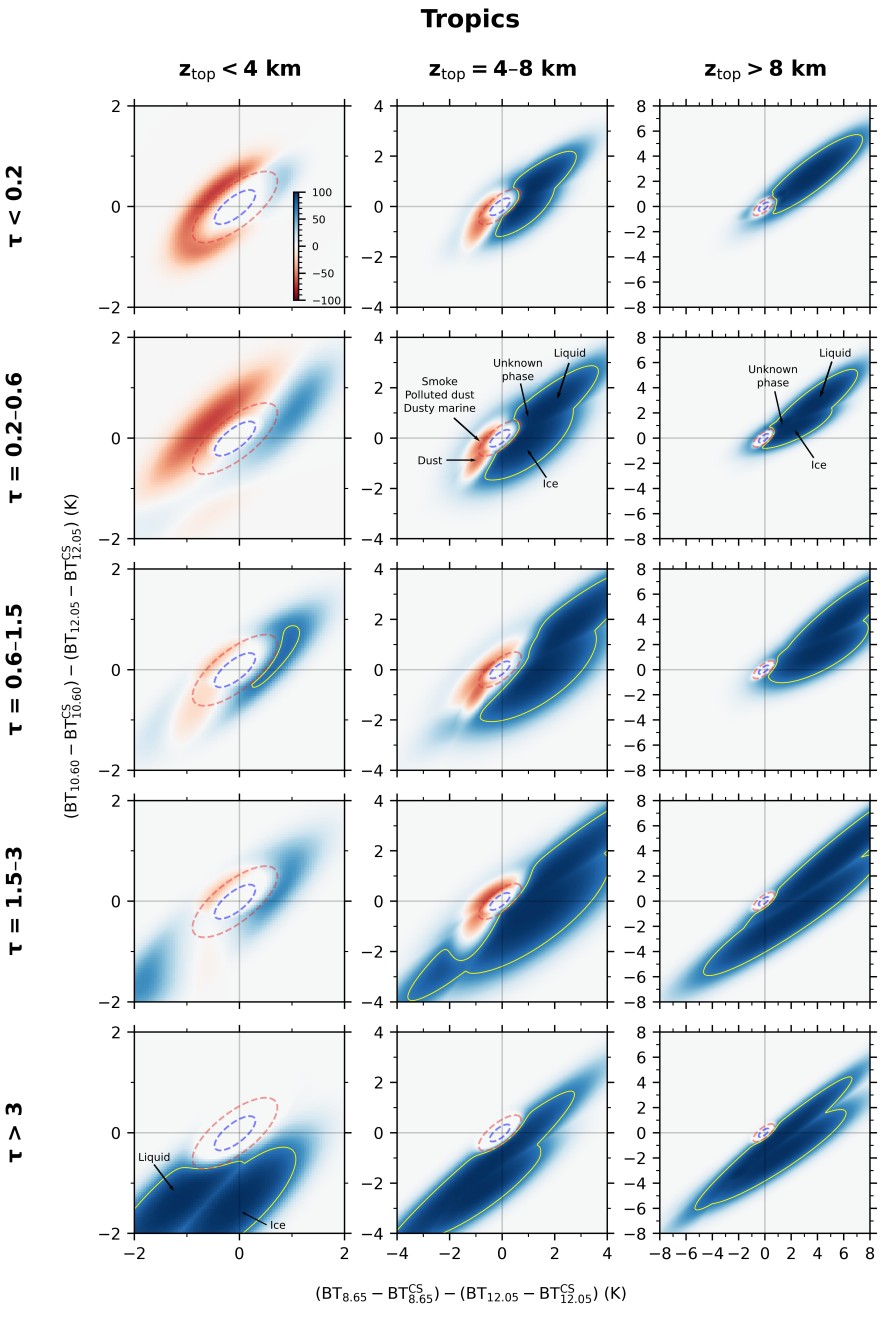

**Figure 10.** IIR CAD score derived from CALIPSO V4 confident observations of all types of cloud and aerosol in the tropics. Negative (red) IIR CAD score correspond to aerosol signatures and positive (blue) IIR CAD score to cloud signatures. Yellow solid lines represent the $|\mathrm{CAD_{score,IIR}}| = 70$ isocontours, separating ambiguous and confident IIR classification. Note the difference in the axis scales for each column. The red and blue dashed ellipses represent the observed 95 % and 50 % confidence intervals of IIR clear-sky observations (see Fig. 2).



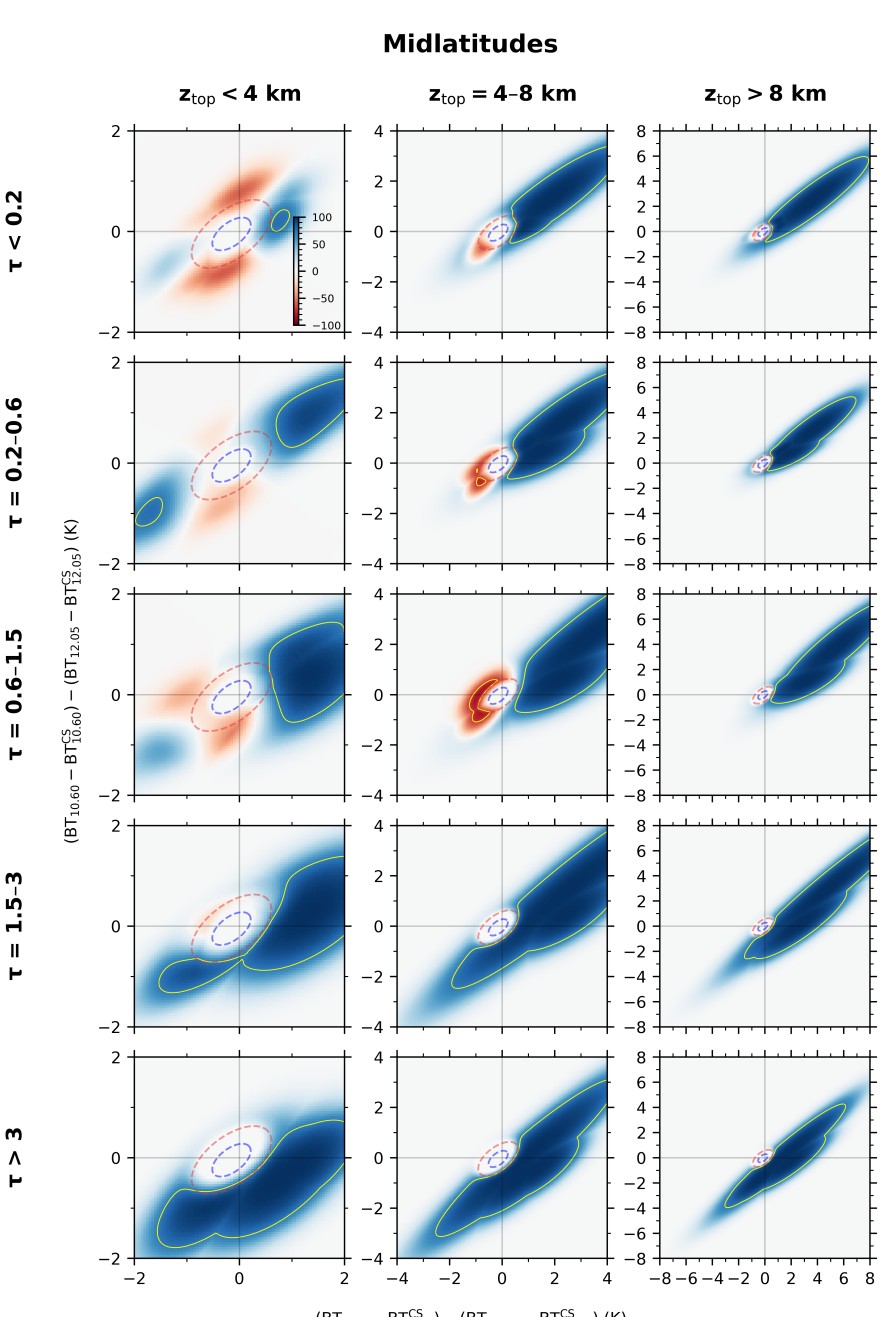

**Figure 11.** As Fig. 10 for the midlatitudes.



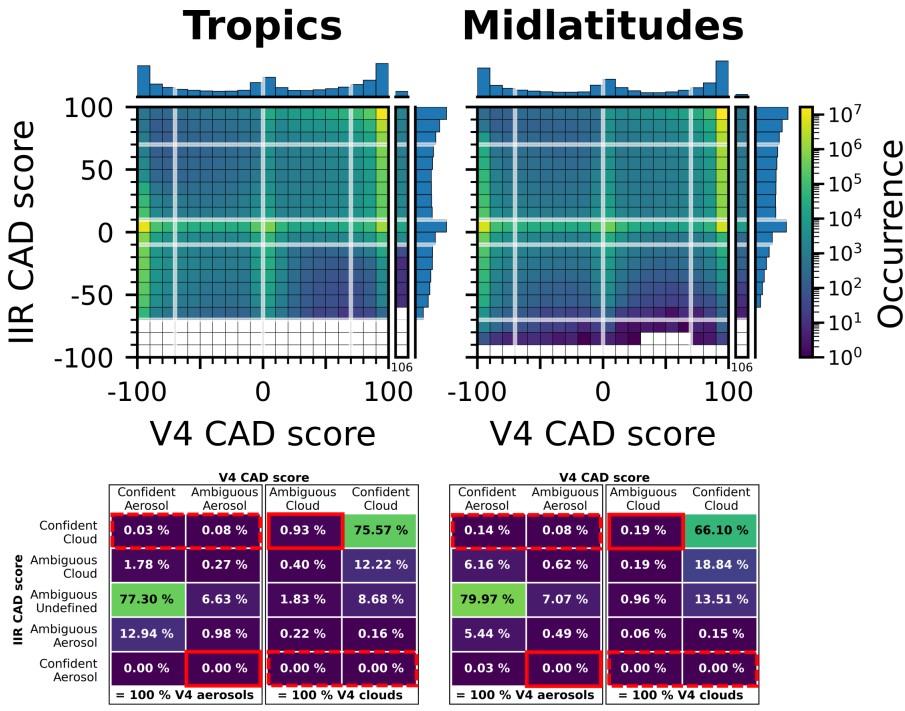

**Figure 12.** IIR CAD score vs V4 CAD score for all monolayer observations for the 2008–2019 period. Transparent white lines show the limit between confident and ambiguous cases. 2-D and 1-D histograms are in log scale. Also shown is the special CAD score 106 ("cirrus fringes") of the V4 classification shown here. Tables below plots provide the fraction of IIR confident ($|\mathrm{CAD}_{\mathrm{score,V4}}| > 70$), ambiguous ($10 < |\mathrm{CAD}_{\mathrm{score,V4}}| < 70$), and undefined ($|\mathrm{CAD}_{\mathrm{score,V4}}| < 10$) classifications of the V4 cloud and aerosol layers. Red rectangles show where the V4 CAD algorithm can benefit from IIR observations to confirm ambiguous cloud or aerosol layers (solid border line) and correct false cloud or aerosol detections (dashed border line).

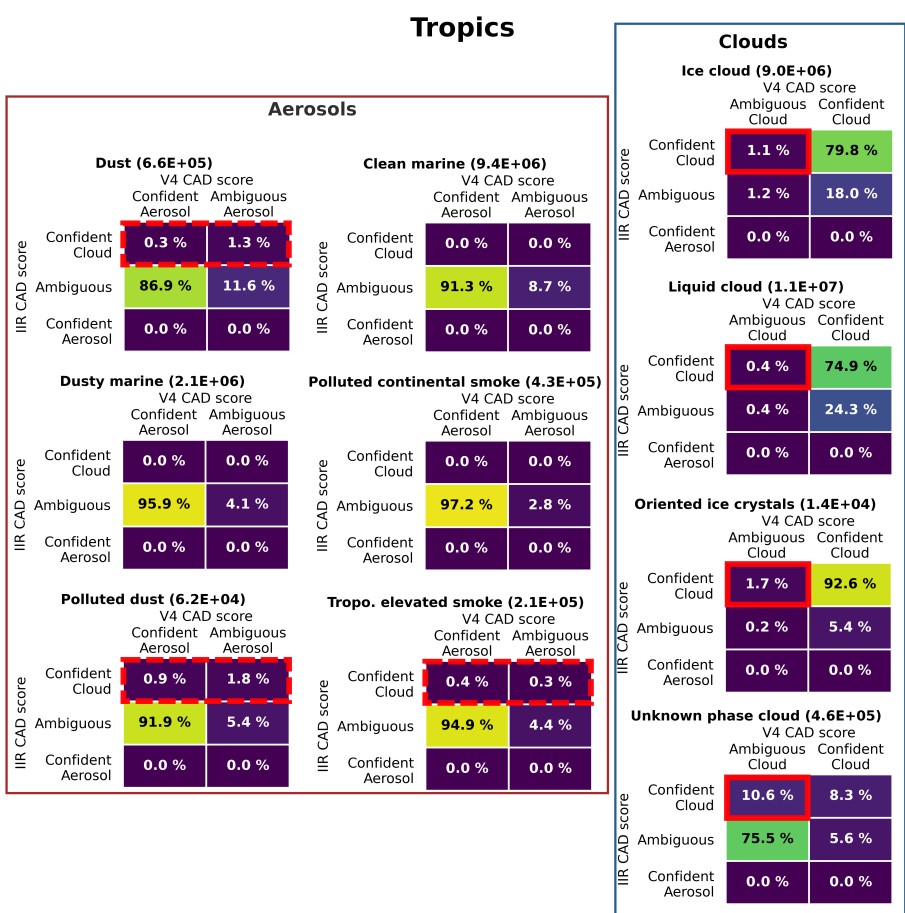

**Figure 13.** Confusion matrices of IIR vs V4 confident and ambiguous monolayer observations for each layer type defined by the V4 algorithm in the tropics for the 2008–2019 period. Red rectangles show where the V4 CAD algorithm can benefit from IIR observations to confirm ambiguous clouds (solid border line) and correct false aerosol detections (dashed border line). Aerosol types with less than 10 000 observations are not shown.



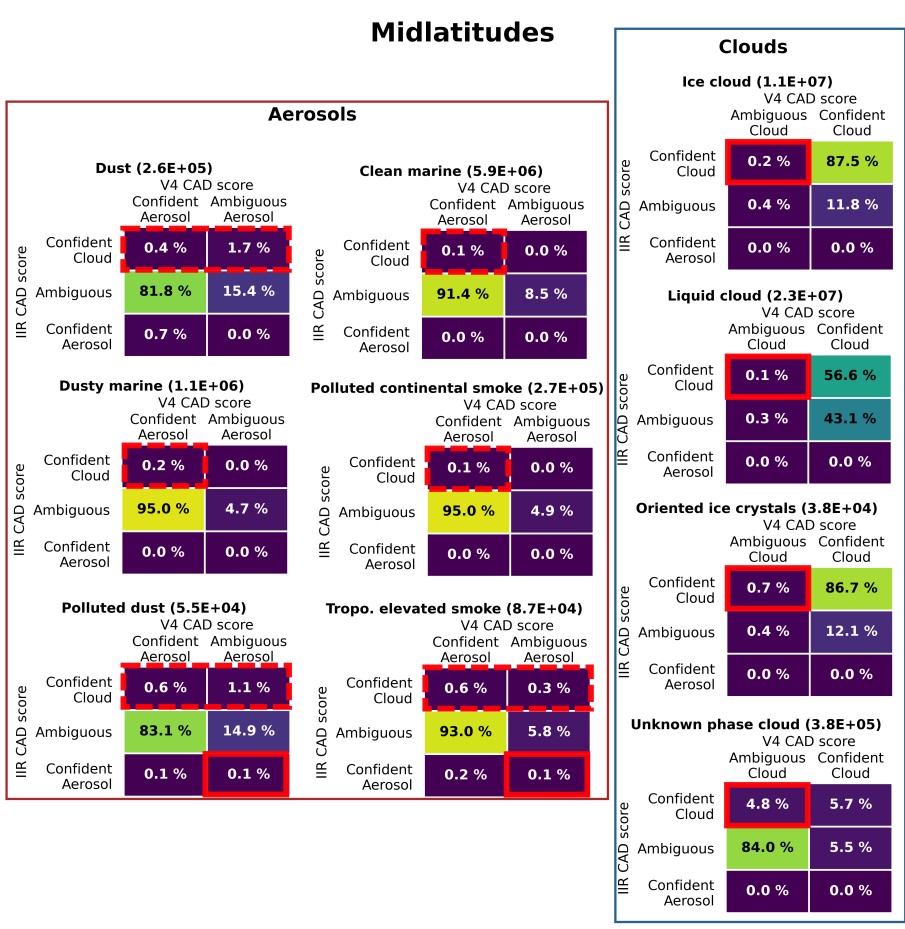

**Figure 14.** As Fig. 13 for the midlatitudes. Red rectangles show where the V4 CAD algorithm can benefit from IIR observations to confirm ambiguous clouds or aerosols (solid border line) and correct false aerosol detections (dashed border line).

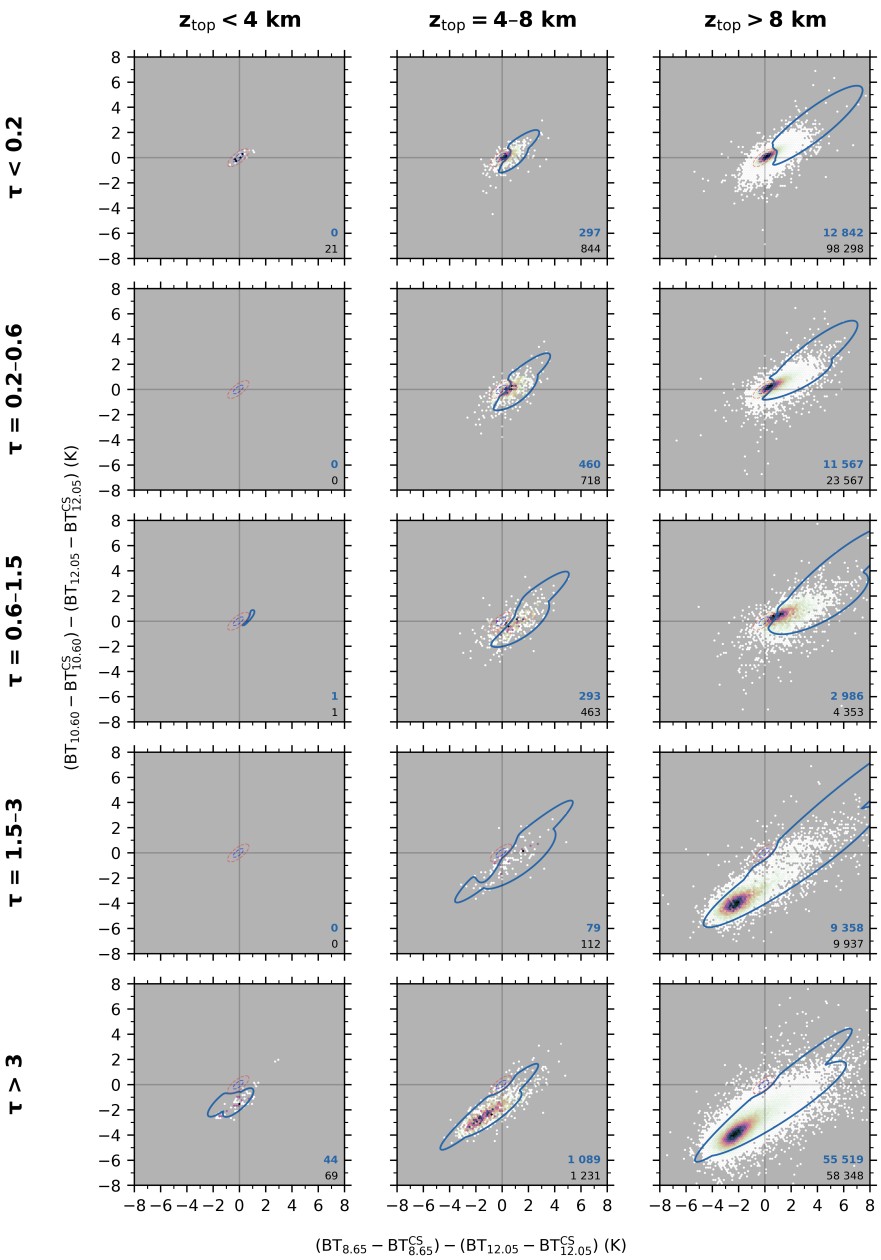

**Figure 15.** IIR signature of the V4 ambiguous ($0 \leq |\mathrm{CAD_{score,V4}}| < 70$) ice cloud observations for the 2008–2019 period in the tropics. IIR signature inside the blue contour ($|\mathrm{CAD_{score,IIR}}| \geq 70$) are confidently classified as cloud by the IIR CAD algorithm. At the bottom-right corner of each subplot are indicated the total number of V4 ambiguous ice cloud observations (black) and those that are confidently classified as cloud by IIR (blue).





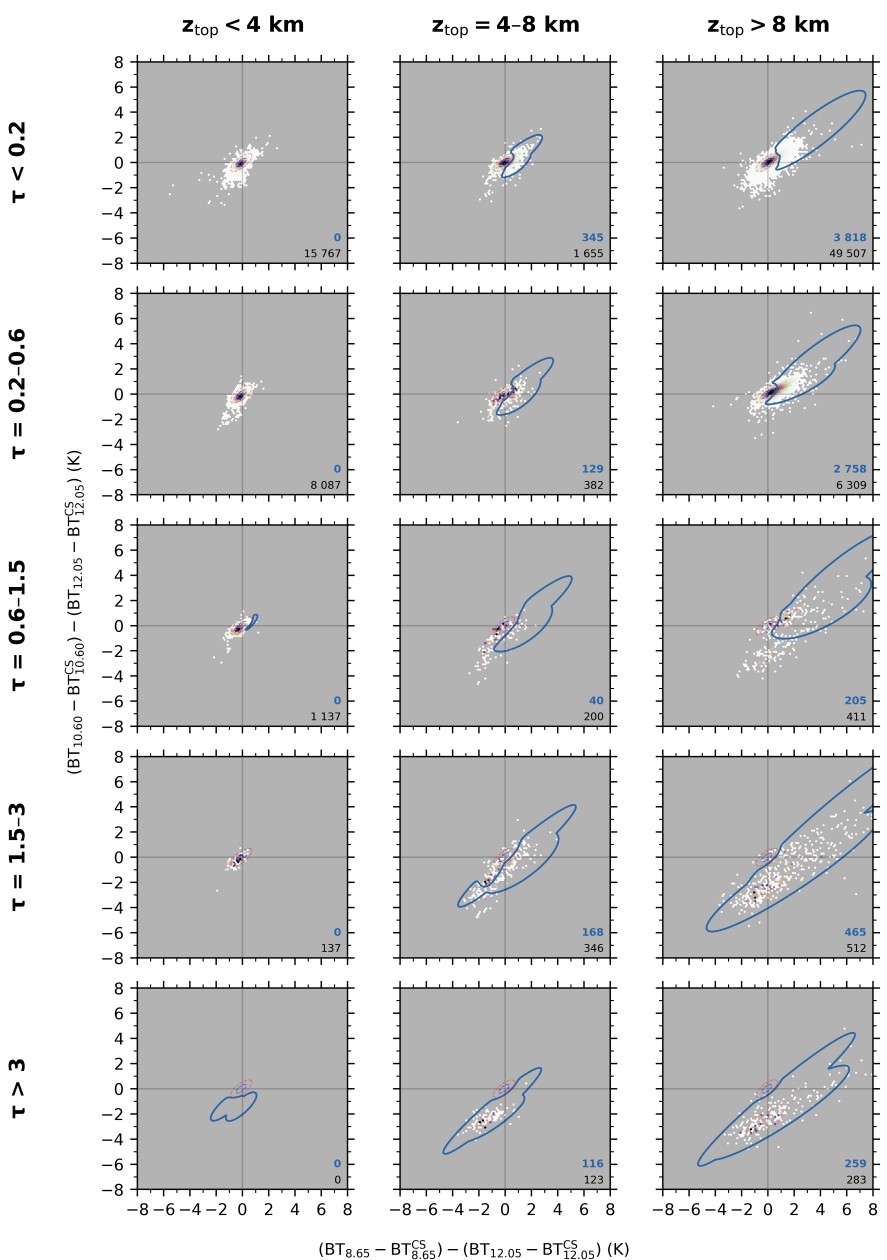

**Figure 16.** As Fig. 15 for dust monolayer observations declared as ambiguous by the V4 algorithm.



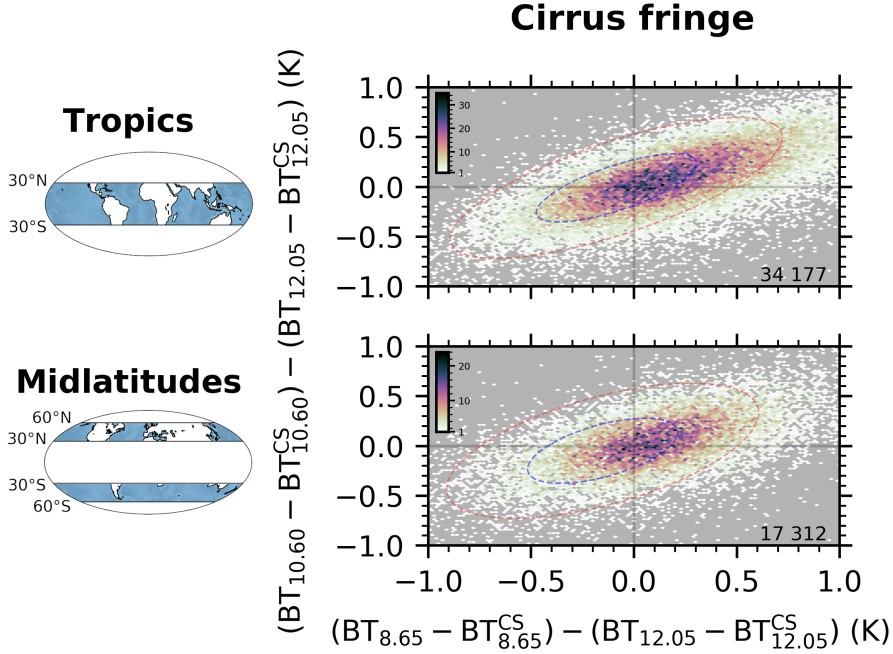

**Figure 17.** As Fig. 5 but for cirrus fringes with V4 special CAD score of 106, i.e., layers detected as aerosols by the V4 CAD algorithm first and then toppled to clouds by the V4 "cirrus fringe amelioration" algorithm (Liu et al., 2019). They mainly are very thin layer (95 % with $\tau < 0.2$) and at high altitude (97 % with $z_{\text{top}} > 8$ km in the tropics and 22 % with $4$ km $< z_{\text{top}} < 8$ km and 78 % with $z_{\text{top}} > 8$ km in the midlatitudes).

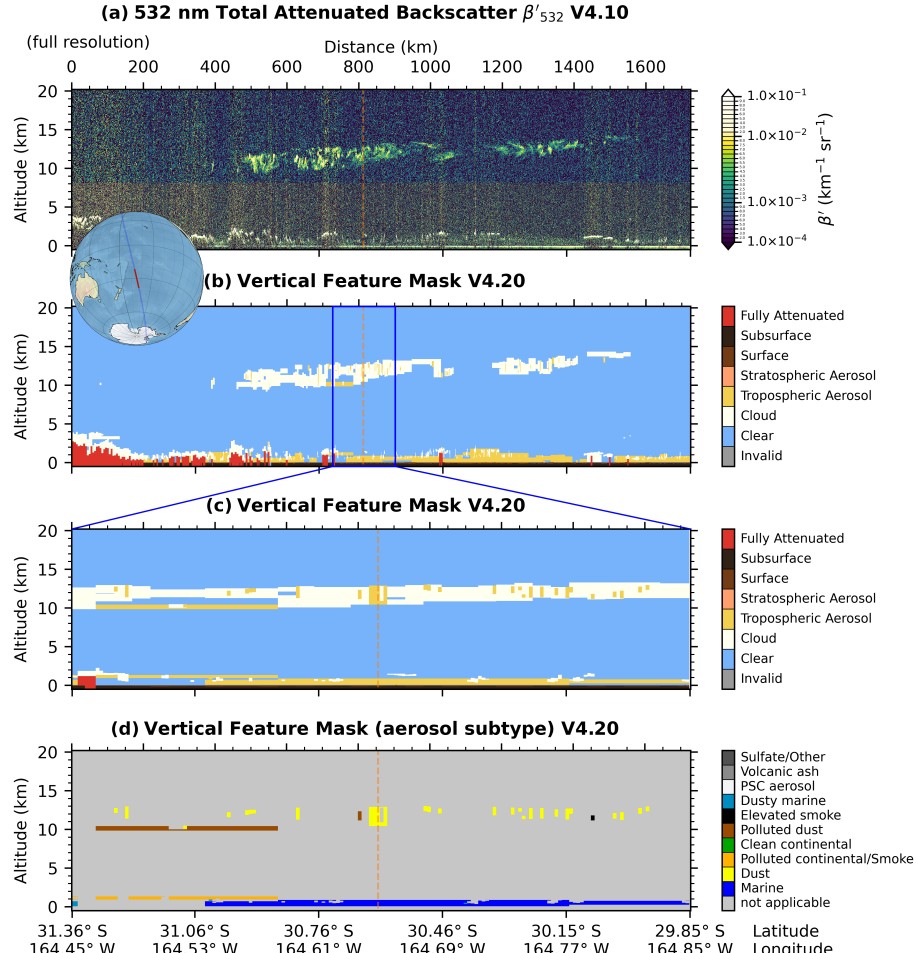

**Figure 18.** Misclassified dust monolayer by V4 CAD algorithm ($\mathrm{CAD_{score,V4}} = -36$) identified as cloud by IIR observation ($\mathrm{CAD_{score,IIR}} = 99$). The misclassified aerosol layer is located between 10.5 and 12.9 km altitude in the 5-km atmospheric column (5-km index #1541) indicated by a dashed line. CALIOP daytime observations on 1 March 2018, 1:01:13–1:05:25 UTC, over the South Pacific: (a) 532 nm total attenuated backscatter V4.10, (b) vertical feature mask V4.20, (c) vertical feature mask zoom around the atmospheric column of interest, and (d) aerosol subtyping. The marine aerosol layer in the planetary boundary layer is detected with 80-km-horizontal averaging and does not affect the IIR observation.



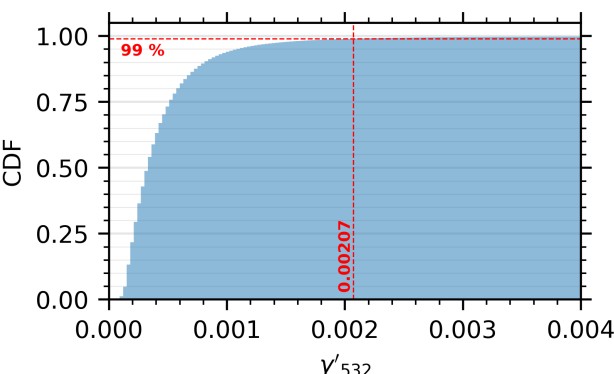

**Figure A1.** Cumulative distribution function of the integrated attenuated backscatter at 532 nm of the uppermost layers detected at 80-km horizontal averaging and with top altitude above 8 km for the 2008 year. $\gamma'_{532} \leq 0.00207$ for 99 % of these layers.

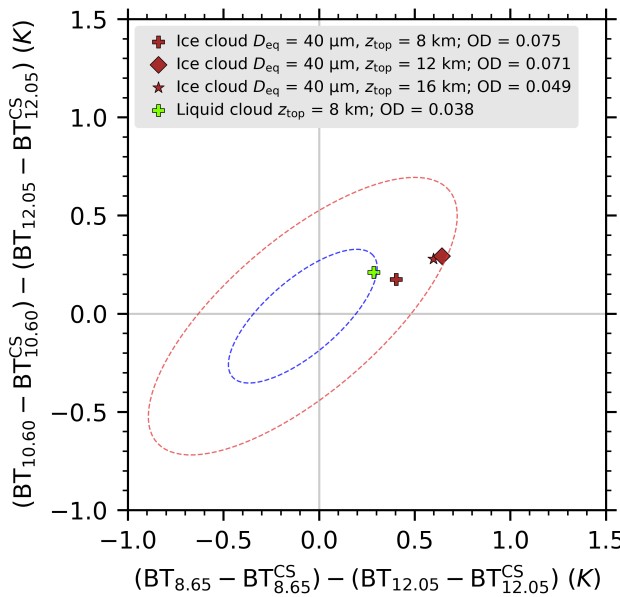

**Figure A2.** Radiative transfer simulation of the IIR signature of layers detected at 80-km horizontal averaging with the largest possible integrated attenuated backscatter. They represent the very worst possible cases that can affect the IIR measurments. We note than even those worst cases do not escape from the clear-sky uncertainty region.





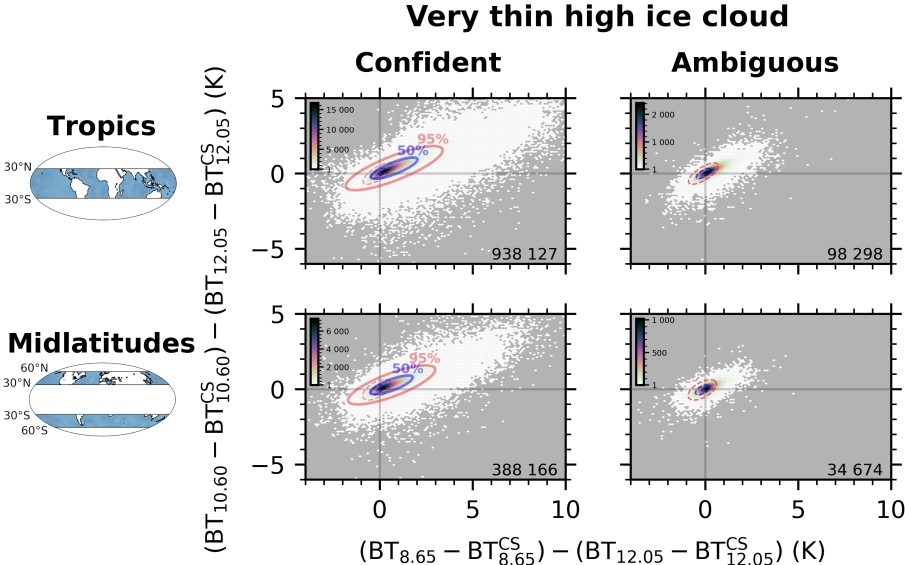

**Figure B1.** As Fig. 5 for thin ice cloud monolayer observations in the upper troposphere ($\tau < 0.2$ and $z_{\mathrm{top}} > 8$ km). The red and blue solid ellipses represent the 95 % and 50 % confidence intervals of the 2-D gaussian PDF estimated from the confident observations of this specific $z_{\mathrm{top}}-\tau$ grid and used in the IIR CAD score definition (Eq. (1)).

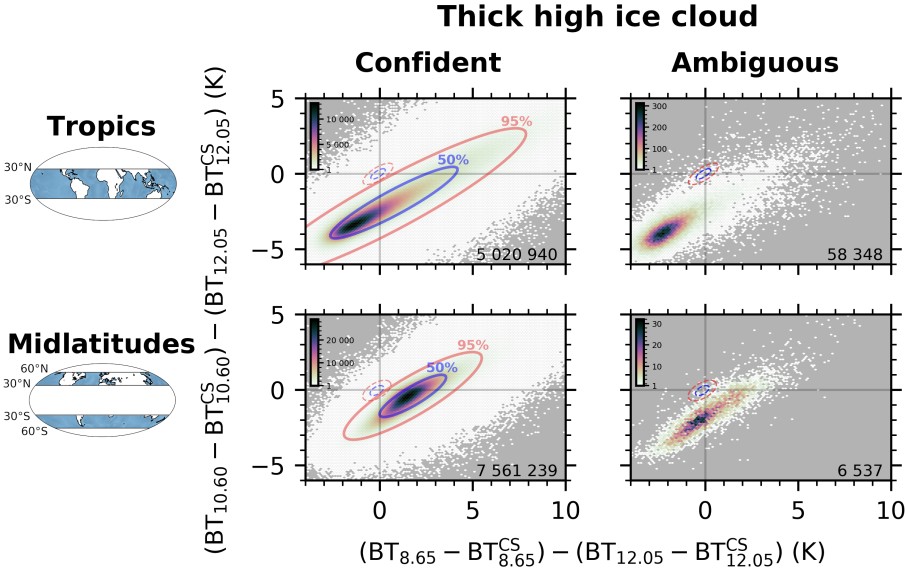

**Figure B2.** As Fig. B1 for thick ice cloud monolayer observations in the upper troposphere ($\tau > 3$ and $z_{\mathrm{top}} > 8$ km).

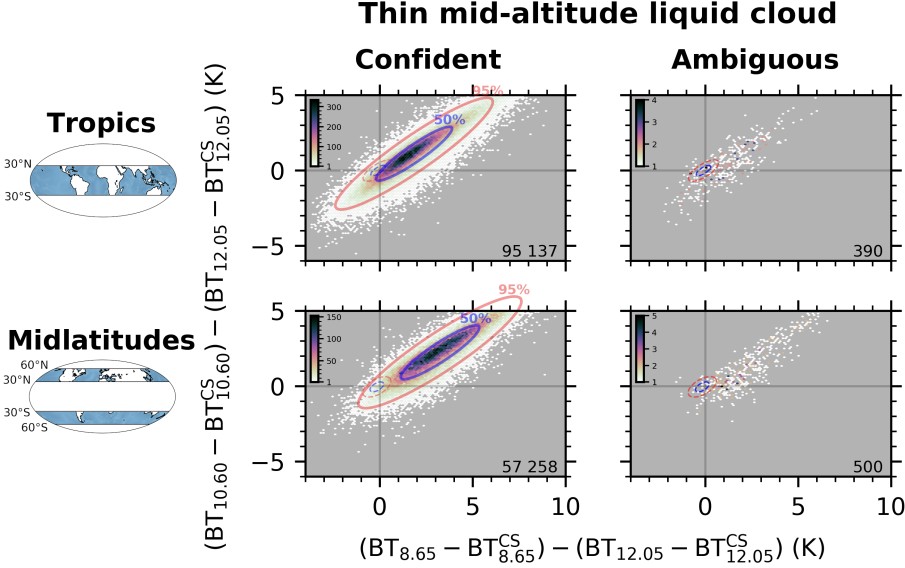

**Figure C1.** As Fig. 6 for liquid cloud monolayer observations with $0.6 < \tau < 1.5$ and $4$ km$< z_{\text{top}} < 8$ km. The red and blue solid ellipses represent the 95 % and 50 % confidence intervals of the 2-D gaussian PDF estimated from the confident observations of this specific $z_{\text{top}}$–$\tau$ grid and used in the IIR CAD score definition (Eq. (1)).

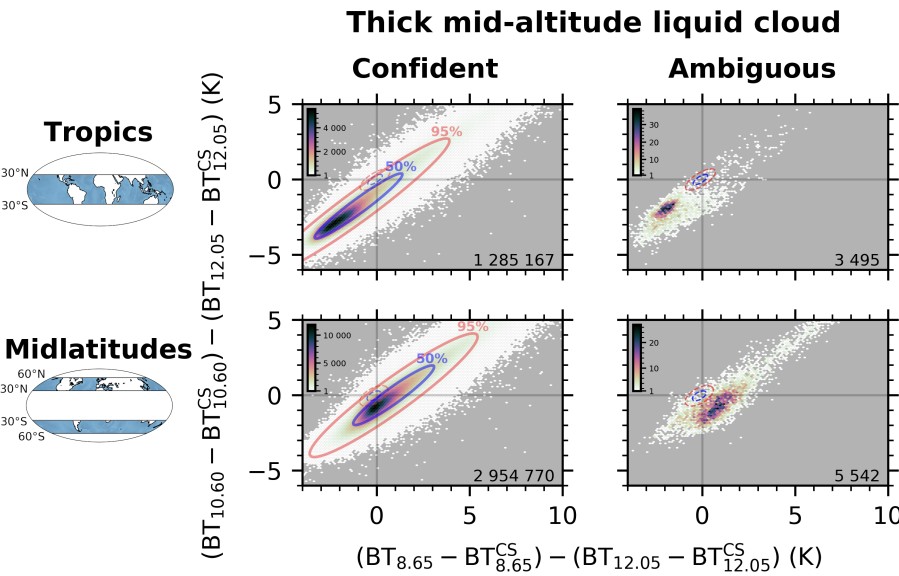

**Figure C2.** As Fig. C1 for liquid cloud monolayer observations with $\tau > 3$ and $4$ km$< z_{\text{top}} < 8$ km.