# Peer review of "Assessing the benefits of Imaging Infrared Radiometer observations to the CALIOP version 4 cloud and aerosol discrimination algorithm"

_Atmospheric Measurement Techniques, 2021_

## Author Comment (AC1)

Point-by-point responses to the reviews.

We thank Bryan A. Baum and the anonymous reviewer for their reviews and valuable comments. The manuscript has been modified according to the reviewers' suggestions. Below we offer specific, item-by-item responses to all the reviewers' comments. Please also find attached a PDF with the track changes.
* * *
**Anonymous Referee #1**

In this paper, the authors demonstrate and quantify to which extent the Imaging Infrared Radiometer (IIR) channels can improve the accuracy of the Cloud and Aerosol Lidar with Orthogonal Polarization version 4 (V4) cloud and aerosol discrimination (CAD) algorithm.

I believe that this paper is a useful contribution to the field of atmospheric science and I recommend publication. The radiative transfer simulations help to understand the meaning of the observations and the strengths and limitations are well explained. The proposed CAD improvement could be seen as minor (sometime as low as 0.1% of the cases). However, as the dataset covers 16 years, it could happen to be a critical contribution to specific analysis. In addition, this study demonstrates well the capabilities of the IIR for feature classifications.

Minor comments:

p.2 line 27-28 The word "mean" is used several times. I do not know what a "layer-mean" is. I would assume that the signal is averaged vertically on the detected layer. I would recommend to define it or remove the jargon.

- **Response:** Yes, "layer-mean" means that the signal is averaged vertically on the detected layer. This term is used in the definitions given in the publication describing the CALIOP CAD algorithm (Liu et al., 2019). To keep consistency with this publication we prefer "layer-mean" than "layer vertical average".

- **Manuscript modification:**

  ➢ In Introduction (l. 26–28): "[…] where dimensions are the layer-mean (layer vertical average) attenuated backscatter at 532 nm, the layer-mean total attenuated color ratio (mean at 1064 nm divided by the mean at 532 nm), the 532 nm layer-mean volume depolarization ratio, the mid-layer altitude, and the latitude.".

p.2 line 38 "V3 CAD" this sentence would lead me to believe that the "V3 CAD" uses the "infrared spectroradiometers" to "[reduce] the frequency of dense dust misclassified…". If it is not the case, I would suggest to rephrase or use a word like "identify" instead of "reduce".

- **Response:** We agree with the reviewer that the sentence is confusing.

- **Manuscript modification:**

  ➢ In Introduction (l. 38–40): "This technique has proven to be useful for the identification of dense dust layers misclassified as cloud in the V3 CAD algorithm using on-board satellite infrared spectroradiometers (Chen et al., 2010; Naeger et al., 2013a, b).".

p.3 line 70 Please add a short explanation to justify the use of 12 years. Specifically, please explain why 2006-2007 are excluded. It will help the reader to understand if there are limitations to extend your findings to the 2006-2007 dataset.

- **Response:** We agree with the reviewer and added a sentence in introduction to explain why we exclude 2006-2007.

- **Manuscript modification:**

  ➢ In Sect. 2.1 (l. 71–76): "Our analysis is based on data above midlatitude and tropical oceans  during the 12 years from 2008 through 2019. From launch until November 2007, the CALIOP viewing angle was fixed at 0.3°, and in this configuration specular reflections from horizontally aligned ice crystals can contribute significantly to the lidar backscatter signals (Avery et al., 2020). On 28 November 2007, the CALIOP viewing angle was permanently

changed to 3° off nadir. To ensure a uniform instrument configuration throughout the study period we have excluded the 2006 and 2007 measurements from our analyses.".

p.5 line 119 "Unfortunately, IIR will not provide any help…" It seems like a strong statement and I would be inclined to not fully believe it. It is definitely not within the scope of this study but complexity is not identical to impossibility. It's your call but I would suggest to weaken this statement.

- **Response:** Thanks for your comment.

- **Manuscript modification:**

  ➢ In Sect. 2.2 (l. 125–126): "For those column types, extracting useful information from the IIR measurements for the cloud and aerosol discrimination is very challenging, so  they are not studied here.".

p. 6 line 139 "*IIR signature*" OK but it took me a little while to figure out that the words in italic were a specific quantity. You may want to think about something here like adding quotes.

- **Response:** Thanks. We added quotes and removed italics to match with the AMT style guidelines.

- **Manuscript modification:**

  ➢ In Sect. 3 (l. 148–149): "Thus, our study is based on the "IIR signature" of the layer, […]".

p. 9 line 218 – 219 "Note that, in agreement with the simulations," My understanding is that in the simulations (several sentences including this one and several figures including Fig. 3), the inputs of the surface and atmospheric properties (especially temperature) are different in the tropic and mid-latitude. I did not see a mention of cloud properties change so if it was done, I missed it. You may want to explicitly state the difference between tropics/mid-latitudes simulations (or point more explicitly to the appropriate reference) in both the text and one figure caption.

- **Response:** Yes, tropical and midlatitudes simulations differ from their surface temperature, temperature atmospheric profile, and humidity atmospheric profile. The data used are those of the standard tropical and midlatitudes atmosphere from McClatchey (1972) as mentioned in Sect. 2.3. We used the same range of cloud properties in both atmospheres. The shape, size distribution and optical properties for ice cloud particles are from Baum et al. (2011) and from Stephens (1979) for liquid cloud particles as mentioned in Sect. 2.3. We added references to the caption of Fig. 3.

- **Manuscript modification:**

  ➢ In Fig. 3 caption: "The ice cloud particle size distribution is a general habit mixed particle distribution (Baum et al. 2011) with an  effective diameter Deq = 90 µm. The liquid cloud particle size distribution is typical of a stratus cloud (Stephens, 1979). Surface and atmospheric profile properties for both the tropics and midlatitudes are from the standard atmospheres of McClatchey (1972).".

p. 14 line 368 "These IIR observations seem to provide independent global evidence…" I prefer the wording you used in the conclusions. This statement here seems a little too strong. I would suggest to rephrase.

- **Response:** We agree with the reviewer.

- **Manuscript modification:**

  ➢ In Sect. 5.2 (l. 384–386): "These IIR observations suggest that the feature type reclassifications made by the CALIOP V4 "cirrus fringe amelioration" algorithm are indeed correct.".

p. 15 line 410 "very good working" I would suggest to mention that it shows consistency

- **Response:** Thanks for your suggestion.

- **Manuscript modification:**
  - In Sect. 5.2 (l. 426–427): "In this study, our use of independent observations acquired by the IIR confirms the overall reliability and consistency of the lidar-only V4 CAD algorithm. ".

  - In Sect. 5.2 (l. 426–427): "In this study, our use of independent observations acquired by the IIR confirms the overall reliability and consistency of the lidar-only V4 CAD algorithm. ".
* * *
**Bryan A. Baum (Referee)**

The IIR has been used in numerous studies as a complementary source for cloud/aerosol products that are simultaneous with the CALIOP products. Given the exceptional record provided by the CALIPSO platform, and the continual improvements in the cloud and aerosol products over the different Versions (currently Version 4), the current study further shows how the IIR can help to reduce some of the (small) remaining ambiguity in the V4 products. The focus is on those V4 cloud/aerosol retrievals that are derived with less confidence. The use of the IIR in this study suggests a significant improvement for ambiguous V4 cloud retrievals but somewhat less improvement for elevated levels of absorbing aerosols. Stated another way, the IIR can be used to reclassify what are currently ambiguous V4 cloud and aerosol retrievals into more appropriate categories. This is a very nice study that is worthy of publication with mostly very minor revisions listed below for the authors to consider.

A few issues came to mind as I was reading this article.

- The abstract mentions dense dust or elevated smoke layers, but smoke is not mentioned again until page 10, Section 4. The radiative transfer calculations shown in Figure 4 only refer to dust. Figures 8 and 9 refer to only dust as well. It would be interesting to see this expanded to include elevated smoke and perhaps volcanic ash. While events that involve smoke and ash are sporadic, they can have a sizable impact for the period in which they are present.

- **Response:** Radiative transfer calculations are mainly used to illustrate the definition of the "IIR signature" and to provide examples for both absorbing and non-absorbing aerosol layers. Since the partitioning of aerosol types into absorbing and non-absorbing layers realistically bounds the capabilities of the IIR, we prefer not to expand the simulation figures to other CALIOP aerosol sub-types. To address the reviewer's concerns, we have added several sentences explaining why we have not sharpened the focus of the study to specifically included smoke and volcanic ash cases.

- **Manuscript modification:**

  - In Section 3.2.2 (l. 201–210): "We acknowledge that these simulations do not represent an exhaustive view of the IIR signatures of clouds and aerosols since we have chosen specific compositions, microphysics, size distributions, and atmospheric profiles. The goal is to illustrate the variation of the IIR signature with altitude and optical depth and ultimately to provide insight into the possibility to discriminate clouds from aerosols. Nonetheless, preliminary ARTDECO simulations for soot aerosols do not show clear evidence that smoke layers exhibit an IIR signature distinguishable from the clear sky pattern (not shown). For volcanic ash, the IIR signature is very sensitive to the $SO_2$ content of the volcanic plume because it greatly affects the 8 μm radiance (not shown). Note that current CALIPSO product only reports volcanic ash in the stratosphere. Indeed, volcanic ash layers in the troposphere cannot be confidently separated from dust layers, as the characteristics of backscattered lidar signal are largely similar for both aerosol types. At present, distinguishing volcanic ash from dust in the troposphere is best accomplished using manual interpretations of the CALIPSO data informed by back trajectories from volcanic eruption positions.".

- With regards to the IIR, how stable has this sensor been over its 16-year life? Are the radiances being monitored and assessed in comparison to other imagers such as MODIS or VIIRS, and if so, are the results available? Every sensor degrades over time in space, so it would be good to know about the IIR's stability over time.

- **Response:** Over the period CALIPSO was flying in the A-train, IIR has been monitored and assessed in comparison to MODIS and SEVIRI and has shown very high stability over time (Garnier et al., 2017, 2018). At that time comparison to MODIS was very relevant due to the large number of coincidence observations of the two instruments flying in the same orbit. Since CALIPSO now flies in the C-train, IIR sensors continues to be monitored in comparison with SEVIRI. This task is performed by ICARE.

- **Manuscript modification:**

➢ In Introduction (l. 43–45): "The IIR swath is 69-km wide, with a pixel size of 1 km, and the center of the IIR swath is by design temporally and spatially co-located with the CALIOP track. IIR calibration has proven to be very stable (Garnier et al., 2017, 2018) allowing detailed and reliable analysis over time.".

- The IIR has been used to support the CALIOP measurements in this and other studies. It might be useful to include a sentence or two to the Conclusions to suggest what sort of measurements would be useful to further reduce the ambiguities that are the focus of this study. Perhaps polarimetric measurements might improve aerosol detection?

▪ **Response:** Yes, we agree with the reviewer that polarimetric measurements might bring new information useful for the discrimination between cloud and aerosol layers. We also think that existing multispectral observations from MODIS collocated with CALIPSO observations could already provide interesting data. Indeed, the visible channels could help in the discrimination between cloud and aerosol layers. We added a sentence in the Conclusions section.

▪ **Manuscript modification:**

➢ In Conclusions (l. 431–434): "To help further in the discrimination between cloud and aerosol layers, the collocated MODIS level 2 MOD04 aerosol data products based on multispectral analysis for identification of smoke or other non-dust aerosol layers may be complementary to the IIR analysis. The collocated polarimetric measurements from POLDER could also improve the cloud–aerosol discrimination.".

- Would be great to see a similar study over the poles in a future article.

▪ **Response:** We agree with the reviewer that such study would be of great interest. However, the uncertainty in the clear-sky brightness temperature calculations over the poles is too large to allow to use our method. This uncertainty seems to be mainly due to difficulties to correctly estimate sea ice emissivity and could also be due to a bad representation of the polar atmospheric properties. This is under investigation.

Minor revisions - these are simply suggestions for your consideration.

Page 1, line 12: "are confirmed thanks to the IIR". Suggest revising this to something like: are reclassified more appropriately through use of the IIR…

▪ **Response:** Thank you for this suggestion.

▪ **Manuscript modification:**

➢ In Abstract (l. 12–13): "28 % and 14 % of the ambiguous V4 cloud classifications are reclassified more appropriately as confident cloud layers through use of the IIR observations in the tropics and in the midlatitudes respectively.".

Page 2, line 28: cloud or aerosols classification —> cloud or aerosol classification

▪ **Response:** Done.

▪ **Manuscript modification:**

➢ In Introduction (l. 28): "The confidence in the cloud or aerosol classification is […]".

Page 2, line 51: impact of faint layers. What is meant by faint? Optically thin? Be more precise.

▪ **Response:** We agree with the reviewer that "optically thin" is more precise.

▪ **Manuscript modification:**

➢ In Introduction (l. 52–53): "The BTDs are analyzed in terms of their departure from computed clear-sky conditions as an attempt to isolate the impact of  optically thin layers.".

Page 5 (and other places): The phrase "in order to simulate" could be more simply stated as "To simulate". The words "in order" do not add anything and it has been drilled into me by a English literature wordsmith. Just a suggestion.

▪ **Response:** Thank you for your comment.

▪ **Manuscript modifications:**

➢ In Sect. 2.3 (l. 130): "To simulate the behavior of […]".

➢ In Sect. 4.1.2 (l. 265): "To do that [..]".

➢ In Sect. 4.2 (l. 274): "The IIR signature PDFs are derived for several $z_{top}$ - τ ranges  that minimize the PDF widths […]".

➢ In Sect. 4.2 (l. 290–291): "In those equations, $P_{CS}$ corresponds to the clear-sky PDF weighted by a coefficient k = 2  to decrease […]".

➢ In Sect. 4.2 (l. 296–297): "A background PDF $P_{bkg}$ = 0.05 is added to the equations  to avoid […]".

➢ In Appendix A (l. 441–442): "To impact the IIR measurement, its emission temperature needs to be […]".

Page 8, line 172: is dense enough - what does this mean? Optically thick? Be more precise.

▪ **Response:** We agree with the reviewer that "optically thick" is more precise.

▪ **Manuscript modification:**

➢ In Sect. 3.2.1 (l. 177–178): "Indeed, if the cloud is optically thick, its emissivity is close to one […]".

Page 8, line 173: if the cloud is high enough - do you mean above the lower levels of the atmosphere where most of the water vapor resides? Above the boundary layer? This could be stated more clearly.

▪ **Response:** Yes, we mean above the levels of the atmosphere where most of the water vapor resides.

▪ **Manuscript modification:**

➢ In Sect. 3.2.1 (l. 178–180): "Moreover, if the cloud top is sufficiently high (> 8 km), i.e. above the lower levels of the atmosphere where most of the water vapor resides, the IIR signature  is weakly impacted by the remaining water vapor above the cloud.".

Page 8, line 174: again, what is meant by "in very dense clouds". Optically thick?

▪ **Response:** Thank for your comment.

▪ **Manuscript modification:**

➢ In Sect. 3.2.1 (l. 180–181): "As a result, the IIR signature in optically thick clouds represents […]".

Page 8, the sentence "Note that for liquid clouds, even optically thick and high clouds can stay in the clear-sky uncertainty region." This is confusing to me for this reason: A midlevel water cloud composed of supercooled liquid droplets often has ice particles falling out of the base, which depletes the water content of the cloud layer. The remaining liquid droplets tend to be very small, and these small droplets tend to scatter light much more than with

larger droplets, leading to high optical depths even in the IR. So the BTD signatures can be quite large. I do not understand how an optically thick and high cloud composed of liquid water droplets can be anywhere near the clear-sky uncertainty region given the lower brightness temperature of the clouds…unless you are only working with brightness temperature differences and not the actual brightness temperature, which would immediately indicate that the measurement is from a cloud and not clear sky conditions.

- **Response:** We agree with the reviewer that the sentence is confusing. In Fig. 3, we show that the IIR signature of a liquid cloud in the tropics with cloud top at 4 km move away from the center to the top-right as its optical thickness increases, at some point (around an optical thickness of 1.5) the IIR signature makes a U-turn and goes back toward the center, crosses the clear-sky uncertainty region, and ends up in the bottom-left quarter. The IIR signature represents the BTD minus the BTD in clear-sky. The IIR signature of upper altitude and large optical thickness liquid cloud layers then stays in the clear-sky uncertainty region. This does not mean that we cannot detect the layer in IIR observations but that our method cannot tell if the layer is a cloud or an aerosol layer. Here we are interested in the discrimination, not the detection. Moreover, such cloud layers are always classified with high confidence with CALIOP observations (see figure below).

- **Manuscript modification:**

  ➢ In Sect. 3.2.1 (l. 185–189): "Note that  liquid clouds with optical thicknesses larger than 2 and at altitudes above 4 km can  fall in the clear-sky uncertainty region. This occurs because the BTDs of these clouds are close to the clear-sky BTDs, which prevents our method from reliably discriminating cloud from aerosols. However, the cloud–aerosol discrimination of those cloud layers is straightforward in the CALIOP V4 CAD algorithm, and they are always classified with high confidence."

[Figure]

Figure: As Fi. 5 for liquid cloud monolayer observations with 4 km < z_top < 8 km and 1.5 < τ < 3.

Page 8, line 186: IIR observations is —> IIR observations are (by the way, this sentence also includes the phrase "in order to")

- **Response:** Done.

- **Manuscript modification:**

  ➢ In Sect. 3.2.2 (l. 195): "The IIR observations  are […]".

Page 10, line 250: close to dust —> close to a dust

- **Response:** Done.

- **Manuscript modification:**

  - In Sect. 4.1.2 (l. 265–266): "To do that, we select an oceanic region close to a dust source […]".

Page 10, lines 252-253: (dense dust layers at higher altitude and/or larger optical depth are virtually impossible) - this statement (in parentheses in the manuscript) does not make sense to me as written.

- **Response:** We agree with the reviewer that this sentence needs to be rewritten.

- **Manuscript modification:**

  - In Sect. 4.1.2 (l. 266–268): "Figure 9 shows dust, ice cloud, and liquid cloud IIR BTD signatures for layers with 4 km < $z_{top}$ < 8 km and 0.2 < τ < 3 (corresponding to z_top - τ ranges where large IIR signatures from dust can be found).".

Page 10, The sentence beginning online 259 needs a bit of reworking. May I suggest: …are derived for several z-tau ranges that minimize the PDF widths so that the IIR signatures are primarily dependent on layer altitude and optical thickness.

- **Response:** Thank you for the suggestion.

- **Manuscript modification:**

  - In Sect. 4.1.2 (l. 274–275): "The IIR signature PDFs are derived for several $z_{top}$ - τ ranges  that minimize the PDF widths because the IIR signatures are  primarily dependent on layer altitude and optical thickness.".

Page 12, line 317-318: …can be confirmed thanks to IIR —> can be reclassified more appropriately with the aid of IIR measurements

- **Response:** Thanks for your comment.

- **Manuscript modification:**

  - In Sect. 5.1 (l. 332–334): "Some of the V4 ambiguous clouds can be reclassified more appropriately with the aid of IIR measurements (28 % in the tropics and 14 % in the midlatitudes) as they received a confident IIR CAD score.".

Page 13, line 341: The primary aerosol mentioned to this point in the article is dust. This is the first mention of elevated smoke. Volcanic aerosols are not mentioned specifically. Perhaps the manuscript could be more clear on what types of aerosol classification will be aided by the IIR. It seems to me that elevated smoke (with soot particles), volcanic ash, and mineral dust are all candidates for the IIR as they can each have significant absorption in the IR.

- **Response:** Dust is by far the aerosol sub-type which is the most affected by the IIR classification. We added a sentence to mention this.

- **Manuscript modification:**

  - In Sect. 5.1 (l. 366–367): "Note that for both the tropics and the midlatitudes, 95 % of the aerosol layers reclassified as cloud layers are dust or polluted dust.".

Page 13, line 352: since IIR —> Since the IIR

- ▪ **Response:** Thank you.
- ▪ **Manuscript modification:**
  - ➢ In Sect. 5.1 (l. 368): "Since the IIR confident cloud CAD score is […]".

Page 15 line 399: show less good agreement —> show lower agreement

- ▪ **Response:** Thank you.
- ▪ **Manuscript modification:**
  - ➢ In Conclusions (l. 415): "Comparison between V4 and IIR CAD scores of V4 aerosols show less goodlower agreement with […]".

Page 15, line 403: observed by CALIOP have shown —> observed by CALIOP was shown

- ▪ **Response:** Thank you.
- ▪ **Manuscript modification:**
  - ➢ In Conclusions (l. 418–419): A specific analysis of a case study of the misclassified dust layers observed by CALIOP have was shown to be consistent with […]".

Page 15, line 407: However, compare to clear-sky —> However, compared to the clearsky

- ▪ **Response:** Thank you.
- ▪ **Manuscript modification:**
  - ➢ In Conclusions (l. 423): However, compared to the clear-sky IIR signature PDF […]".

Page 15, line 410: very good working —> very good skill

- ▪ **Response:** This sentence has been rephrased. Please see comment from anonymous reviewer #1.